# The unbiased estimation of the fraction of variance explained by a model

**Dean A. Pospisil** [1]*, **Wyeth Bair** [1,2,3,4]

**1** Department of Biological Structure, University of Washington, Seattle, Washington, United States of America, **2** Washington National Primate Research Center, University of Washington, Seattle, Washington, United States of America, **3** University of Washington Institute for Neuroengineering, Seattle, Washington, United States of America, **4** Computational Neuroscience Center, University of Washington, Seattle, Washington, United States of America

* deanp3@uw.edu

## Abstract

The correlation coefficient squared, $r^2$, is commonly used to validate quantitative models on neural data, yet it is biased by trial-to-trial variability: as trial-to-trial variability increases, measured correlation to a model's predictions decreases. As a result, models that perfectly explain neural tuning can appear to perform poorly. Many solutions to this problem have been proposed, but no consensus has been reached on which is the least biased estimator. Some currently used methods substantially overestimate model fit, and the utility of even the best performing methods is limited by the lack of confidence intervals and asymptotic analysis. We provide a new estimator, $\hat{r}^2_{ER}$, that outperforms all prior estimators in our testing, and we provide confidence intervals and asymptotic guarantees. We apply our estimator to a variety of neural data to validate its utility. We find that neural noise is often so great that confidence intervals of the estimator cover the entire possible range of values ([0, 1]), preventing meaningful evaluation of the quality of a model's predictions. This leads us to propose the use of the signal-to-noise ratio (SNR) as a quality metric for making quantitative comparisons across neural recordings. Analyzing a variety of neural data sets, we find that up to ∼ 40% of some state-of-the-art neural recordings do not pass even a liberal SNR criterion. Moving toward more reliable estimates of correlation, and quantitatively comparing quality across recording modalities and data sets, will be critical to accelerating progress in modeling biological phenomena.

## Author summary

Quantifying the similarity between a model and noisy data is fundamental to advancing the scientific understanding of biological phenomena, and it is particularly relevant to modeling neuronal responses. A ubiquitous metric of similarity is the correlation coefficient, but this metric depends on a variety of factors that are irrelevant to the similarity between a model and data. While neuroscientists have recognized this problem and proposed corrected methods, no consensus has been reached as to which are effective. Prior methods have wide variation in their accuracy, and even the most successful methods lack

**Data Availability Statement:** The MT data is available at: http://www.neuralsignal.org/data/21/nsa2021.1.html V4 data is available at: https://www.kaggle.com/c/uwndc19. The code to calculate all estimators and confidence intervals is

available at: https://github.com/deanpospisil/er_est.

**Funding:** This work was supported by a National Science Foundation Graduate Research Fellowship DGE-1256082 (D.A.P.), National Institutes of Health Grant NEI R01-EY02999, and National Institutes of Health Grant NEI R01-EY027023 (W. B.). All funders had no role in study design, data collection and analysis, decision to publish, or preparation of the manuscript.

**Competing interests:** The authors have declared that no competing interests exist.

confidence intervals, leaving uncertainty about the reliability of any particular estimate. We address these issues by developing a new estimator along with an associated confidence interval that outperforms all prior methods. We also demonstrate how a signal-to-noise ratio can be used to usefully threshold and compare noisy experimental data across studies and recording paradigms.

## Introduction

Building an understanding of the nervous system requires the quantification of model performance on neural data, and this often involves computing Pearson's correlation coefficient between model predictions and neural responses. Yet this typical estimator, $\hat{r}^2$, is fundamentally confounded by the trial-to-trial variability of neural responses: a low $\hat{r}^2$ could be the result of a poor model or high neuronal variability.

One approach to this problem is to average over many repeated trials of the same stimulus in order to reduce the influence of trial-to-trial variability. With a finite number of trials, this approach will never wholly remove the influence of noise and its confounding effect, moreover, the collection of additional trials is expensive. A more principled approach has been to account for trial-to-trial variability in the estimation of the fraction of explainable variance or $r^2$. Most often, this takes the form of attempting to estimate what the $r^2$ would have been in the absence of trial-to-trial variability. Here we call this quantity $r_{ER}^2$, the $r^2$ between the model prediction and the expected response (ER) of the neuron (i.e., the 'true' mean, or expected value, of the estimated tuning curve). While a variety of solutions have been proposed to estimate this quantity [1–10], they have not been quantitatively compared, thus there is no basis to reach a consensus on which methods are appropriate, or more importantly inappropriate. We find that several estimators still in recent use have large biases. Estimators that did have relatively small biases lacked associated confidence intervals, thus the degree of uncertainty in these sometimes highly variable estimates remains ambiguous. Finally, none of these methods have been analyzed asymptotically to give a theoretical guarantee that they will converge to $r_{ER}^2$, i.e., it has not been shown that they are consistent estimators.

To address these substantial problems, we introduce $\hat{r}_{ER}^2$, which is a simple analytic estimator of $r_{ER}^2$, along with a method for generating $\alpha$-level confidence intervals. We validate our estimator in simulation, prove that it is consistent, and provide head-to-head comparisons to prior methods. We then demonstrate the use of $\hat{r}_{ER}^2$ and its confidence interval on two sets of neural data. We find many cases where neuronal data is so noisy that estimates of $r_{ER}^2$ provide little inferential power about the quality of a model fit. This naturally leads to a useful metric of the quality of a neuronal recording that we will refer to as the signal-to-noise ratio (SNR), and which can be interpreted in terms of the number of trials needed to reliably detect tuning. Across a diverse set of neural recordings, we find that many neurons do not pass even a liberal criterion for providing meaningful insight into the quality of a model fit.

## Results

Our results are organized as follows. First, we give the essential intuition into the source of the bias in $\hat{r}^2$ and we explain how $\hat{r}_{ER}^2$ removes this bias. Next, we evaluate $\hat{r}_{ER}^2$ through simulation and compare it to prior methods. We then demonstrate the method on two neural data sets: one from a study of motion direction tuning in area MT and one from a study of responses to

natural images in area V4. Finally, we develop an estimator, $\widehat{SNR}$, based on the signal-to-noise ratio (SNR), as a metric to determine the inferential power of a given neuronal recording.

## Bias of $\hat{r}^2$ and its correction

Consider a typical scenario in sensory neuroscience where the responses of a neuron to $m$ stimuli across $n$ repeated trials of each stimulus have been collected and the average of these responses, the estimated tuning curve (Fig 1, dashed green line), is compared to responses predicted by a model (red line). These responses could be spike counts from a neuron but could just as well be any other neural signal. Even if the $m$ expected values of the neuronal response, $\mu_i$ (solid green trace), perfectly correlate with the model predictions, $v_i$ (red trace is scaled and shifted relative to green), the $m$ sample averages, $Y_i$ (dashed green trace), will deviate from their expected value owing to the sample mean's variability. Here, we quantify this variability using the variance, $\sigma^2$, of the distribution of responses from trial-to-trial (see Methods, "Assumptions and terminology for derivation of unbiased estimators"). We assume $\sigma^2$ is constant across responses to different stimuli, which can be achieved by applying a variance stabilizing transform to the data. The variance of the sample mean for all stimuli will thus be $\frac{\sigma^2}{n}$. Owing to the variance of the sample mean, the reported $\hat{r}^2$ can be appreciably less than 1 even though the $r^2$ between the underlying expected values of the neuronal response and the model is 1.

The quantity we attempt to estimate in this paper is $r^2$ between the model predictions ($v_i$) and the expected neuronal responses ($\mu_i$). We will call this quantity $r^2_{\mathrm{ER}}$, the fraction of variance of the 'expected response' explained by the model:

$$r^2_{\mathrm{ER}} = \frac{\left(\sum_{i=1}^{m}(v_i - \bar{v})(\mu_i - \bar{\mu})\right)^2}{\sum_{i=1}^{m}(v_i - \bar{v})^2 \sum_{i=1}^{m}(\mu_i - \bar{\mu})^2}. \tag{1}$$

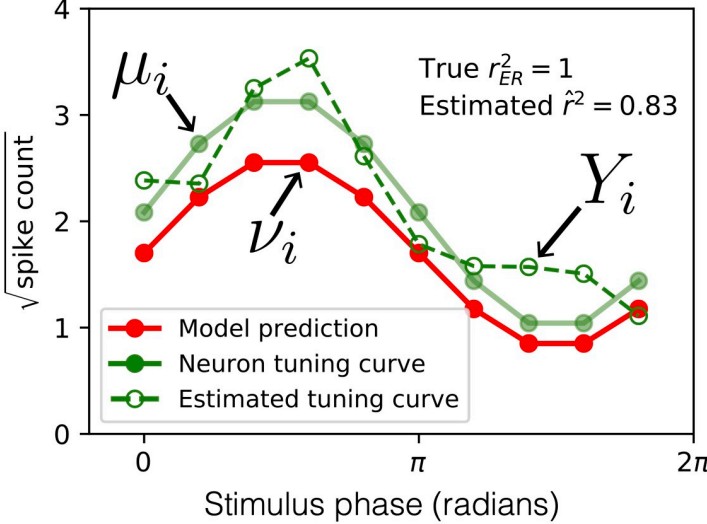

**Fig 1. Sampling noise confounds estimation of the correlation between model prediction and neuronal tuning curve.** The expected (true) spike counts in response to a set of 10 stimuli (solid green points) is perfectly correlated with a model (red points), yet owing to sampling error (neural trial-to-trial variability) the estimated tuning curve (green open circles) has correlation less than one with the model ($\hat{r}^2 = 0.83$).

We will show that the naive sample estimator, which uses $Y_i$ in place of $\mu_i$,

$$\hat{r}^2 = \frac{\left(\sum_{i=1}^m (v_i - \bar{v})(Y_i - \bar{Y})\right)^2}{\sum_{i=1}^m (v_i - \bar{v})^2 \sum_{i=1}^m (Y_i - \bar{Y})^2}, \tag{2}$$

has an expected value that can be well approximated as the ratio of the expected values of its numerator and denominator as follows (for asymptotic justification see Methods, "Inconsistency of $\hat{r}^2$ in $m$"):

$$
\begin{aligned}
\mathrm{E}[\hat{r}^2] &\approx \frac{\mathrm{E}\left[\left(\sum_{i=1}^m (v_i - \bar{v})(Y_i - \bar{Y})\right)^2\right]}{\mathrm{E}\left[\sum_{i=1}^m (v_i - \bar{v})^2 \sum_{i=1}^m (Y_i - \bar{Y})^2\right]} \\
&= \frac{\left(\sum_{i=1}^m (v_i - \bar{v})(\mu_i - \bar{\mu})\right)^2 + \frac{\sigma^2}{n}\sum_{i=1}^m (v_i - \bar{v})^2}{\sum_{i=1}^m (v_i - \bar{v})^2 \sum_{i=1}^m (\mu_i - \bar{\mu})^2 + \frac{\sigma^2}{n}(m-1)\sum_{i=1}^m (v_i - \bar{v})^2}.
\end{aligned}
\tag{3}
$$

While the terms on the left in the numerator and denominator are the same as $r_{\mathrm{ER}}^2$, the terms on the right are proportional to the trial-to-trial variability ($\sigma^2$) and cause $\hat{r}^2$ to deviate from $r_{\mathrm{ER}}^2$. This is the essential problem: $\hat{r}^2$ is biased away from $r_{\mathrm{ER}}^2$ by terms proportional to the amount of variability, $\frac{\sigma^2}{n}$, in the estimated responses.

The strategy we take to solve this problem is straightforward: find unbiased estimators of these noise terms and subtract them from the numerator and denominator of Eq 2 for $\hat{r}^2$, thus:

$$\hat{r}_{\mathrm{ER}}^2 = \frac{\left(\sum_{i=1}^m (v_i - \bar{v})(Y_i - \bar{Y})\right)^2 - \frac{\hat{\sigma}^2}{n}\sum_{i=1}^m (v_i - \bar{v})^2}{\sum_{i=1}^m (v_i - \bar{v})^2 \sum_{i=1}^m (Y_i - \bar{Y})^2 - \frac{\hat{\sigma}^2}{n}(m-1)\sum_{i=1}^m (v_i - \bar{v})^2}, \tag{4}$$

where $\hat{\sigma}^2$ is an unbiased estimator for trial-to-trial variability, after a variance stabilizing transform if necessary. Typically $\hat{\sigma}^2 = \hat{s}^2$, the sample variance, but not necessarily. For example if stimuli are shown only once ($n = 1$), then an assumed value of trial-to-trial variability could be substituted into $\hat{\sigma}^2$. The numerator and denominator of the fraction $\hat{r}_{\mathrm{ER}}^2$ are unbiased estimators of the numerator and denominator of $r_{\mathrm{ER}}^2$; therefore, this solution is approximate since the expected value of a ratio is not necessarily the ratio of the expected values of the numerator and denominator (see Methods, "Bias of $\hat{r}_{\mathrm{ER}}^2$"). Yet we show in simulation that the approximation is very good for typical neural statistics, and we show analytically that, unlike $\hat{r}^2$, our estimator $\hat{r}_{\mathrm{ER}}^2$ converges to the true $r_{\mathrm{ER}}^2$ as the number of stimuli $m \to \infty$ (see Methods, "Consistency of $\hat{r}_{\mathrm{ER}}^2$ in $m$"). We next evaluate this estimator in simulation.

## Validation of $\hat{r}_{\mathrm{ER}}^2$ by simulation

To demonstrate the effectiveness and key properties of $\hat{r}_{\mathrm{ER}}^2$, we ran a simulation with $m = 362$ stimuli, $n = 4$ repeats, and $\sigma^2 = 0.25$ (the trial-to-trial variance of Poisson neuronal response after a variance-stabilizing transform, see Methods: "Assumptions and terminology for derivation of unbiased estimator"). This corresponds, for example, to a minimal experiment to characterize shape tuning in V4 neurons, which requires hundreds of unique shapes and takes on the order of 1 hour [2]. In the case where the model prediction ($v_i$) and expected response ($\mu_i$) were perfectly correlated (as in Fig 1) and SNR was moderate at 0.5, the distribution of the naive estimator, $\hat{r}^2$, is centered well below 1 (Fig 2A, blue). Thus, the model appears to be a poor fit to data that it in fact generated, indicating that $\hat{r}^2$ is a poor estimator of $r_{\mathrm{ER}}^2$. On the other hand, the distribution of our corrected estimator, $\hat{r}_{\mathrm{ER}}^2$, is appropriately centered at 1 (Fig 2A, orange). Approximately 50% of the time our estimator exceeds 1, taking on impossible values of $r_{\mathrm{ER}}^2 \in [0, 1]$, but this is necessary to achieve unbiased estimates for high $r_{\mathrm{ER}}^2$ because truncating the values would shift the mean below 1.

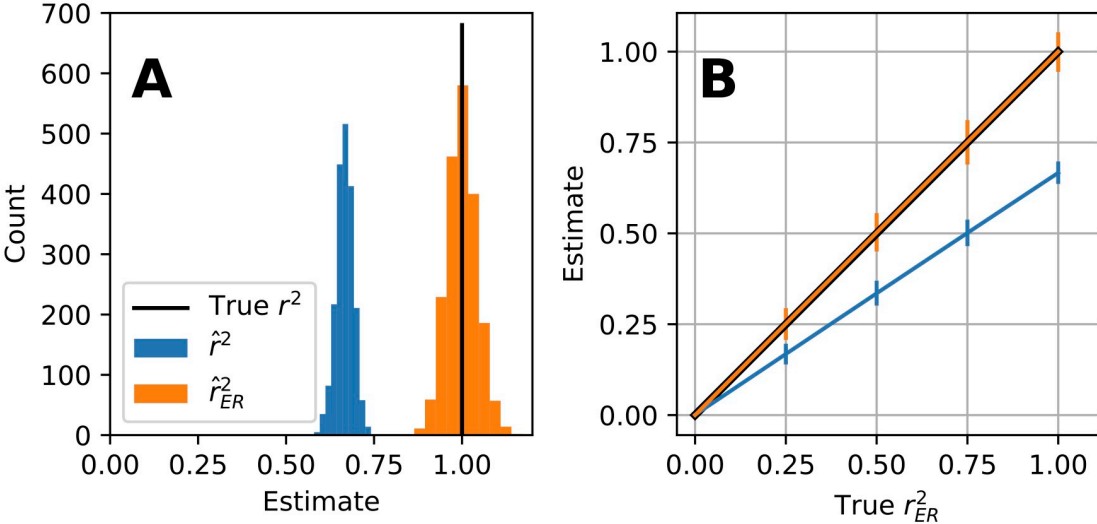

**Fig 2. Simulation of the naive $\hat{r}^2$ and unbiased $\hat{r}^2_{\text{ER}}$ estimators for model-to-neuron fits at varying levels of $r^2_{\text{ER}}$ where $m = 362$, $n = 4$, and $\sigma^2 = 0.25$. (A)** For true $r^2 = 1$, at a moderately low SNR = 0.5, $\hat{r}^2$ (blue) is on average 0.67 whereas $\hat{r}^2_{\text{ER}}$ (orange) is on average 1.00. The bias of $r^2_{\text{ER}}$ (see Methods, "Bias of $\hat{r}^2_{\text{ER}}$") is small relative to its variability (90% quantile = [0.93, 1.07] vertical bars) and to the bias of $\hat{r}^2$. **(B)** Same simulation as A but at five levels of $r^2_{\text{ER}}$ (0, 0.25, 0.5, 0.75, 1). Lines show mean values of $\hat{r}^2$ (blue) and $\hat{r}^2_{\text{ER}}$ (orange). Black line (beneath orange) shows true $r^2_{\text{ER}}$; error bars show 90% quantile.

We evaluated the estimators $\hat{r}^2$ and $\hat{r}^2_{\text{ER}}$ at five values of $r^2_{\text{ER}}$ (0, 0.25, 0.5, 0.75, 1) and plotted the mean and 90% quantiles. Fig 2B shows that $\hat{r}^2$ (blue line) grossly underestimates $r^2_{\text{ER}}$ (black line) at all levels except for $r^2_{\text{ER}} = 0$, whereas $\hat{r}^2_{\text{ER}}$ (orange line) correctly estimates the true correlation $r^2_{\text{ER}}$ (orange and black lines overlap). Thus the estimator $\hat{r}^2_{\text{ER}}$ performs favorably in this simulation. We ran similar simulations on the square root of Poisson distributed spikes counts and found similar results for both low and high average firing rates. Next, we characterize the performance of $\hat{r}^2_{\text{ER}}$ relative to $\hat{r}^2$ in simulations that cover a wide range of the parameters, $m$, $n$ and SNR.

## Asymptotic properties of $\hat{r}^2_{\text{ER}}$ and $\hat{r}^2$

We ran simulations to determine the bias and variance of $\hat{r}^2_{\text{ER}}$ relative to $\hat{r}^2$ as a function of the parameters SNR, $n$, and $m$. Fig 3A shows that as SNR increases, $\hat{r}^2$ (blue) and $\hat{r}^2_{\text{ER}}$ (orange) converge ($r^2_{\text{ER}} = 0.75$, $n = 4$, $m = 362$). Thus, for neuronal recordings where variation in response strength across stimuli is large relative to trial-to-trial variability, these two estimators should have similar values. At low values of SNR, e.g., 0.1, $\hat{r}^2$ has a large downward bias (mean $\hat{r}^2 = 0.23$), whereas $\hat{r}^2_{\text{ER}}$ has a small upward bias relative to its own variability and to the bias of $\hat{r}^2$ (for the source of this bias see Methods, "Bias of $\hat{r}^2_{\text{ER}}$"). This small upward bias of $\hat{r}^2_{\text{ER}}$ quickly diminishes as SNR increases, whereas the large negative bias of $\hat{r}^2$ remains across a much wider range of SNR. The essential problem this simulation reveals is that if SNR varies widely from neuron to neuron, the bias in the naive estimate will cause apparent variation in $r^2$ across neurons that depends on SNR and not on the underlying tuning curve. Neuronal SNR is not typically under experimental control, making this problem difficult to avoid.

The number of repeats, $n$, is under the experimenter's control but is expensive to increase. Fig 3B shows how $\hat{r}^2$ and $\hat{r}^2_{\text{ER}}$ converge as $n$ increases. Thus the bias in $\hat{r}^2$ can be reduced by increasing the number of repeats, but to achieve this requires a very high number of repeats

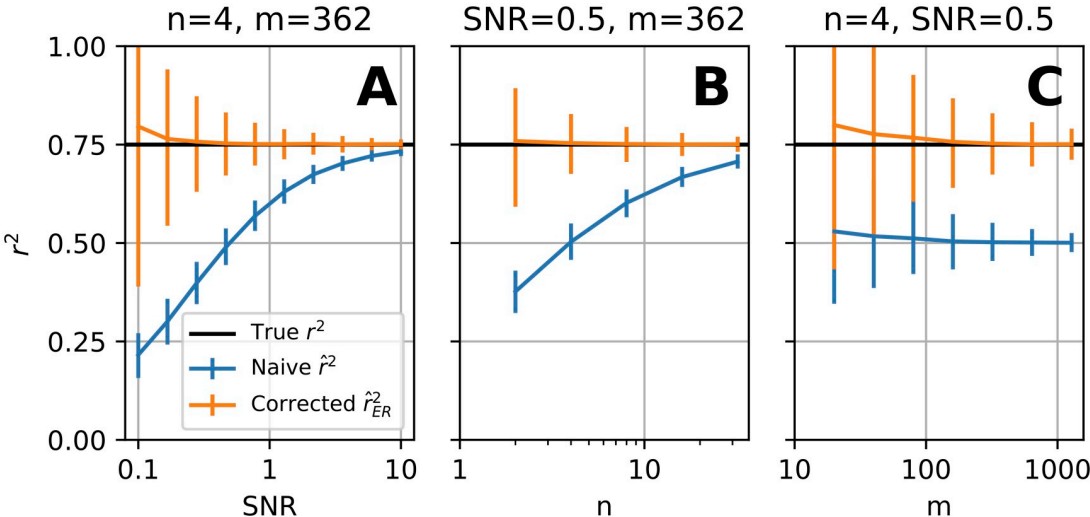

**Fig 3. Comparison of $\hat{r}^2$ and $\hat{r}^2_{ER}$ for estimating model-to-neuron fit across broad, relevant ranges of SNR, *n*, and *m*. (A)**
Average performance of naive $\hat{r}^2$ (blue) and corrected $\hat{r}^2_{ER}$ (orange) as a function of SNR for a simulation where true $r^2_{ER} = 0.75$
(horizontal black line), $m = 362$, $n = 4$, and $\sigma^2 = 0.25$. Error bars indicate 90% quantiles. **(B)** Performance of estimators as a
function of *n*, the number of repeats of each stimulus. Simulation like (A), except SNR = 0.5 and *n* is varied. **(C)** Performance as a
function of *m*, the number of unique stimuli, for a low number of repeats ($n = 4$). Like (A), except SNR = 0.5 and *m* is varied.

for low SNR. An advantage of $\hat{r}^2_{ER}$ is that even for low *n*, it on average estimates the true corre-
lation to the model (orange trace overlaps black trace, Fig 3B), providing a large gain in total
trial efficiency for estimating the quality of model fit.

When increasing the number of stimuli, *m*, unlike the previous two cases, $\hat{r}^2$ and $\hat{r}^2_{ER}$ do *not*
converge to the same value (Fig 3C). While variability of both estimators decreases (90% quan-
tiles narrow), it is clear in simulation that $\hat{r}^2$ is not a consistent estimator of $r^2_{ER}$ in *m* since it
does not converge to $r^2_{ER} = 0.75$. While there is a small upward bias of $\hat{r}^2_{ER}$ for low *m*, as *m*
increases this bias is reduced (see Methods, "Consistency of $\hat{r}^2_{ER}$ in *m*").

## Comparison to prior methods

Accounting for noise when interpreting the fit of models to neural data has been examined
and applied in the neuroscientific literature for some time [1–10]. Several studies have fol-
lowed the approach of attempting to estimate the upper bound on the quality of fit of a model
given noise and then referencing the measure of fit to this quantity. Roddey et al. [1] estimate
this upper bound by computing their estimate of model fit, 'coherence', across split trials then
plotting coherence of the data to the model predictions relative to the split trial coherence.
Yamins et al. [7] normalize $r^2$ with split-trial correlation transformed by the Spearman-Brown
prediction formula (averaged across randomly resampled subsets of trials); we will call this
$\hat{r}^2_{\text{norm-split-SB}}$. Hsu et al. [4] also use split-half correlation (averaged across randomly resampled
subsets of trials), to estimate an upper bound ($CC_{max}$) by a transformation they derive attempt-
ing to estimate the correlation of the true mean with the firing rate of the neuron. For purpose
of comparison, we square this estimator and call it $CC^2_{\text{norm-split}}$. Schoppe et al. [8] improve upon
this method by giving an analytic form, thus removing the need for resampling. They do this
by using the 'signal-power' (SP) estimate developed by Sahani and Linden [3], thus we call
their estimator $CC^2_{\text{norm-SP}}$. Kindel et al. [10] take inspiration from Schoppe et al., except to

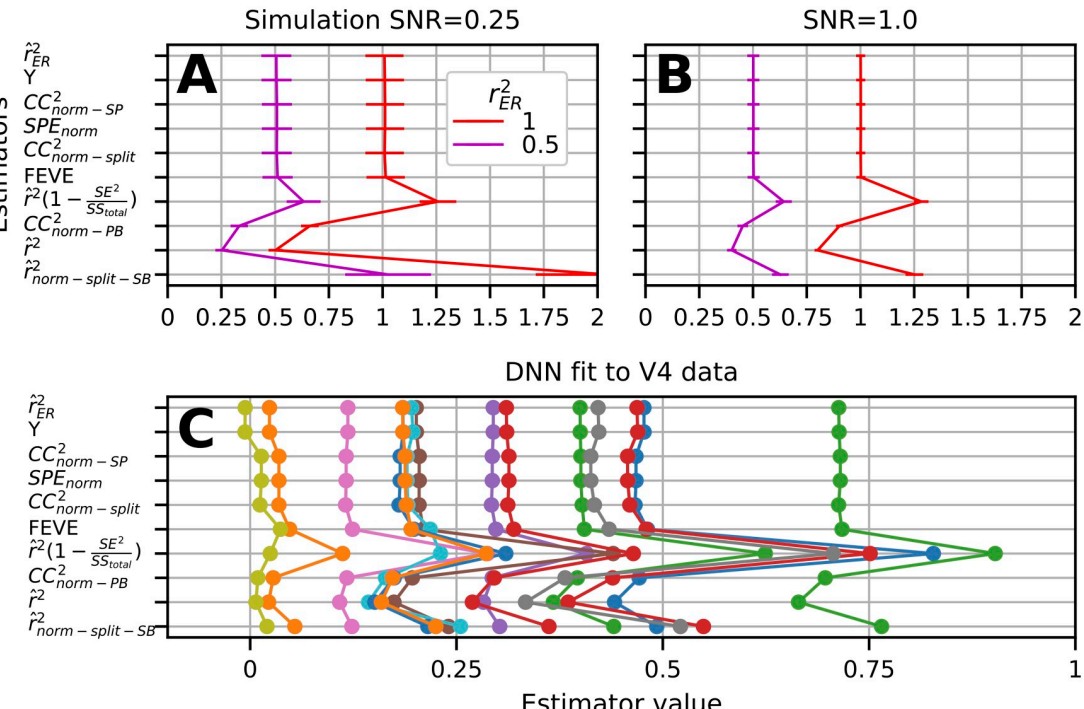

**Fig 4. Comparison of $\hat{r}^2_{ER}$ with published estimators of $r^2_{ER}$ on the basis of simulated and real data. (A)** Low SNR (0.25) simulation where estimators on vertical axis are sorted from top to bottom by smallest MSE with respect to estimating $r^2_{ER} = 1$. Traces show mean and SD of each estimator. **(B)** Same simulation at higher SNR (1.0) but same $m$, $n$. **(C)** Estimated fit of DNN to V4 data by $\hat{r}^2_{ER}$ and published estimators. Each trace is the estimated fit of the model for one neural recording.

estimate $CC_{max}$ they measure the correlation of responses from a Gaussian simulation (based on the sample mean and variance of the neural data) with the sample mean. We square their estimator and call it $CC^2_{norm\text{-}PB}$ (PB for parametric bootstrap). Pasupathy and Connor [2] estimate the fraction of total variance accounted for by trial-to-trial variability, intuitively the fraction of unexplainable variance, then use it to normalize $\hat{r}^2$. We call this estimator $\hat{r}^2(1 - \frac{SE^2}{SS_{total}})$. With a similar motivation, Cadena et al. [9] provide a metric they call "fraction explainable variance explained" (FEVE). They form the ratio of mean squared prediction error over total variance of the response (except subtracting off an estimate of trial-to-trial variability from both) and subtract this ratio from one. While all of these methods might be intuitively appealing, the quantities to which they converge, and their relationship to $r^2_{ER}$ is unclear.

Unlike the above approaches, we follow a line of research [3, 6] that explicitly attempts to construct an unbiased estimator of $r^2$ in the absence of noise (see Methods, "Prior analytic methods of estimating $r^2_{ER}$"). Heretofore many of the methods reviewed above have not been quantitatively validated and none have been directly compared. We now compare all these methods with respect to estimating $r^2_{ER}$. We exclude from this comparison David and Gallant [5] because their method depends on a large number of repeated trials, at which point the estimators' utility decreases.

We quantified the ability of all methods to estimate $r^2_{ER}$ in a simulation with $n = 4$ trials and $m = 362$ stimuli (see Methods, "Simulation procedure"). We sort the estimators (Fig 4, y-axis) by their MSE in a test case where $r^2_{ER} = 1$. We generally find, $\hat{r}^2_{ER}$, $\Upsilon$, $SPE_{norm}$, $CC^2_{norm\text{-}SP}$, FEVE, and $CC^2_{norm\text{-}split}$ are all comparable in their performance (red trace, top 6 points) with $\hat{r}^2_{ER}$

performing slightly, but significantly, better. $SPE_{norm}$ and $CC^2_{norm-SP}$ are numerically identical in their performance and their trial-to-trial results. On the other hand, $\hat{r}^2(1 - \frac{SE^2}{SS_{total}})$ and $\hat{r}^2_{norm-split-SB}$ both over estimate $r^2_{ER}$, and the naive estimator $\hat{r}^2$, as expected, yields an under estimate (mean = 0.50). In addition $CC^2_{norm-PB}$ underestimates $r^2_{ER}$. When the true $r^2_{ER}$ is 0.5, we find similar results, where $\hat{r}^2(1 - \frac{SE^2}{SS_{total}})$ and $\hat{r}^2_{norm-split-SB}$ produce overestimates (0.63 and 1.04 on average, respectively) and the mean $\hat{r}^2$ is 0.25. Thus, serious caution should be taken when interpreting these last two estimators. We applied these estimators to neural data from V4 fit by a deep neural network (see Methods, "Electrophysiological data") and found a similar pattern of results where the top 6 estimators give similar estimates to each other, $\hat{r}^2(1 - \frac{SE^2}{SS_{total}})$ and $\hat{r}^2_{norm-split-SB}$ tend to be greater than these estimators, and the estimators $\hat{r}^2$ and $CC^2_{norm-PB}$ are lower (Fig 4C). We conclude $\hat{r}^2_{ER}$ is as good as any estimator of $r^2_{ER}$ available, has a simple analytic form, and in contrast to $\Upsilon$, can still be calculated without calculating the sample variance, for example, if no repeats are collected and variance must be assumed (see Discussion, "Relationship to prior methods"). None of the top five prior estimators we reviewed have associated confidence intervals, and thus we now provide confidence intervals for $\hat{r}^2_{ER}$.

## Confidence intervals for $\hat{r}^2_{ER}$

In order to interpret point estimates such as $\hat{r}^2_{ER}$, it is important to be able to meaningfully quantify uncertainty about the estimate relative to the true parameter $r^2_{ER}$. An $\alpha$-level confidence interval (CI) provides an interval that will contain the true parameter $\alpha \times 100\%$ of the time for IID estimates. We considered three typical generic approaches to forming CIs for $\hat{r}^2_{ER}$: the non-parametric bootstrap, the parametric bootstrap, and $BC_a$ [11]. We found all methods to be lacking because they did not achieve the desired $\alpha$ in simulations with ground truth. Motivated by these problems, we developed a novel Bayesian method. We first recount the issues we found with the bootstrap methods and then provide a basic account of the Bayesian method we use throughout the paper. For more detailed exposition, see Methods: "Quantifying uncertainty in the estimator".

The non-parametric bootstrap is a commonly used method to approximate CIs. In our case, it involves randomly re-sampling with replacement from the $n$ trials in response to each of the $m$ stimuli then calculating $\hat{r}^{2(b)}_{ER}$ for the bootstrap sample. Repeating this many times allows the quantiles of the bootstrap distribution of $\hat{r}^{2(b)}_{ER}$ to be used as CIs. We applied this method across a simulated population of 3000 neurons with $m = 40$ and $n = 4$ and found it suffered from two problems. First, the CIs were not centered around $r^2_{ER}$, specifically the interval was too low (Fig 5A), with the upper and lower bounds of the interval (orange and blue traces, respectively) almost always falling below the true value (green). Secondly, as the true $r^2_{ER}$ increased from 0 to 1, CIs contained $r^2_{ER}$ at rates far lower than the desired $\alpha = 0.8$ (Fig 6 cyan trace, open-circles under the trace indicate a significant difference, $p < 0.01$ Bonferoni corrected z-test). Thus at practically all levels of correlation, the non-parametric bootstrap performs poorly. The problem is a result of the expected value of the empirical distribution (the sample mean) being typically much lower than $r^2_{ER}$. To overcome this, we turned to the parametric bootstrap where we could explicitly estimate $r^2_{ER}$ with our estimator $\hat{r}^2_{ER}$. This method approximates CIs by estimating the parameters of an assumed distribution from which samples are generated. In our case it involves estimating $\sigma^2$, $r^2_{ER}$, and the variance of the neuronal tuning curve $d^2$ (see Results, "Signal-to-noise ratio as recording quality metric") and then simulating observations from the distribution with these parameters to calculate $\hat{r}^{2(PB)}_{ER}$.

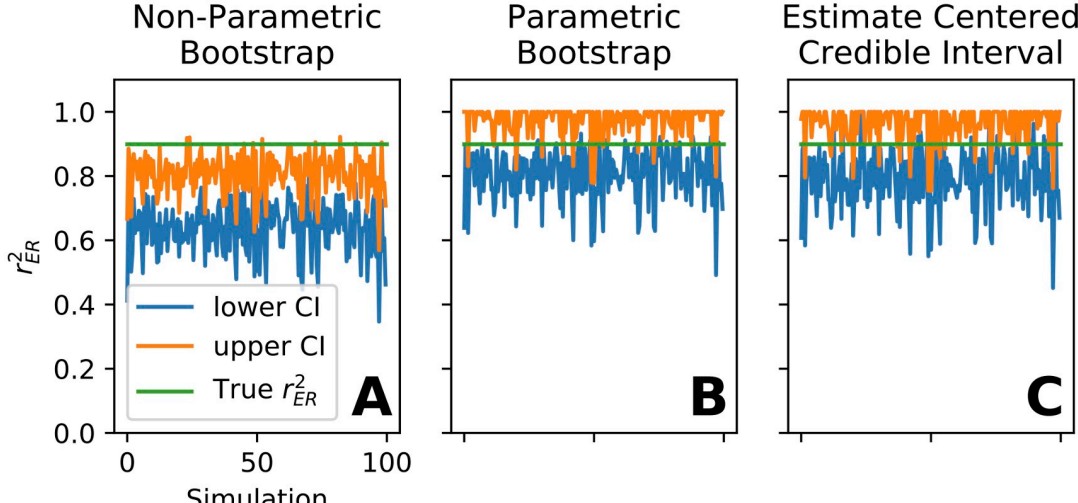

**Fig 5. Validation of confidence interval (CI) methods by simulation—example CIs for three methods.** Simulation parameters: $n = 4$, $m = 40$, true $r^2_{ER} = 0.91$, dynamic range $d^2 = 0.25$, trial-to-trial variability $\sigma^2 = 0.25$, and target confidence level $\alpha = 0.8$. Of 2000 independent simulations, CIs for the first 100 are plotted here for three different methods. CIs for all methods were calculated using the same set of randomly generated responses. **(A)** For the non-parametric bootstrap method, the upper end (orange) and lower end (blue) of the CI were almost always both below the true correlation value (0.91, green line), indicating an overwhelming failure to achieve 80% containment of the true value. **(B)** The parametric bootstrap method and **(C)** our ECCI method perform substantially better. Performance of all three methods over the full range of true $r^2_{ER}$ is plotted in Fig 6.

Drawing many $\hat{r}^{2(PB)}_{ER}$ we again use the sample quantiles as CI estimates. Fig 5B shows that this overcomes the main failure of the non-parametric bootstrap, but this method tended to be too conservative for low $r^2_{ER}$ values (Fig 6 red trace below 0.8 at left side) and too liberal for high values (red trace above 0.8 at right side). Deviations such as these are well known for bootstrap percentile methods when the variance is a non-constant function of the mean and/or the distribution of the estimator is skewed [11]. The correction to the bootstrap, the bias-corrected and accelerated bootstrap ($BC_a$), can help ameliorate these issues by implicitly approximating the skewness and the mean-variance relationship from bootstrap samples. We employed $BC_a$ with our parametric bootstrap and found that performance improved relative to the parametric bootstrap (Fig 6 green trace closer to desired $\alpha$ than red for low $r^2_{ER}$) but still deviated from the desired $\alpha$ for low and high $r^2_{ER}$.

We aimed to create a CI with better $\alpha$-level performance. To do this, we assumed uninformative priors on the parameters $\sigma^2$ and $d^2$ so that, conditioned on estimates of these parameters, we can draw from the distribution of $\hat{r}^2_{ER}|r^2_{ER}$ for an arbitrary $r^2_{ER}$ (see Methods, "Confidence Intervals for $\hat{r}^2_{ER}$"). This allows us to compute the highest true $r^2_{ER}$ that would have given an observed $\hat{r}^2_{ER}$ or a lower value in $\alpha/2$ proportion of IID samples. We take this as the high end, $r^2_{ER(h)}$, of our CI. Similarly we determine the low end, $r^2_{ER(l)}$, of the CI as the lowest $r^2_{ER}$ that produces a value greater than or equal to $\hat{r}^2_{ER}$ in $\alpha/2$ of the samples. In Methods we give conditions under which this procedure will provide $\alpha$-level CIs (see Methods, "Confidence intervals for $\hat{r}^2_{ER}$"). In our simulations, this method consistently achieves the desired $\alpha$ at all levels of $r^2_{ER}$ (Fig 6, orange trace). We use this CI method, which we call the estimate-centered credible interval (ECCI), throughout the rest of the paper.

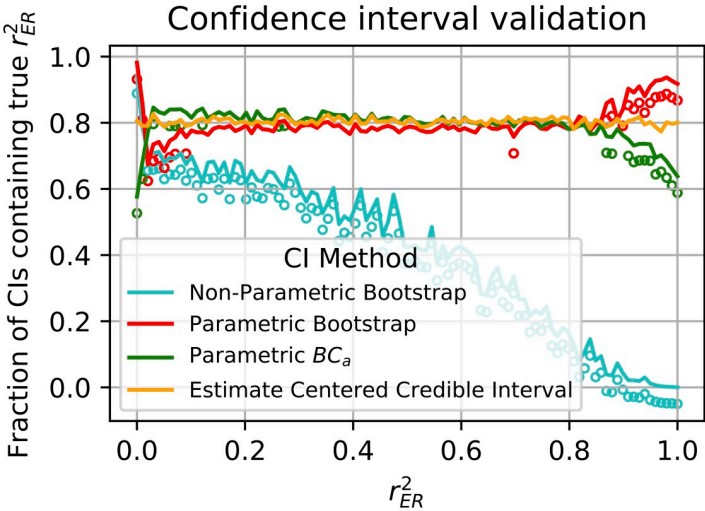

**Fig 6. Comparison of four methods for computing confidence intervals for $\hat{r}^2_{\mathrm{ER}}$ spanning the full range of true correlation.** The fraction of times the CI contained the true value is plotted for each method (see line style inset) as a function of the true correlation value, $r^2_{\mathrm{ER}}$, at 100 values linearly spaced between 0 and 1. The target $\alpha$-level was 0.8. Open circles indicate that the fraction deviated from 0.8 significantly ($p < 0.01$, Bonferroni corrected).

## Application of estimator to MT data

We have shown in simulation that the use of $\hat{r}^2$ introduces ambiguity as to whether a low correlation value was the result of trial-to-trial variability or a poor model, whereas $\hat{r}^2_{\mathrm{ER}}$ removes this ambiguity. Here we demonstrate in neural data how this, in tandem with confidence intervals, allows investigators to distinguish between units that systematically deviate from model predictions versus those that simply have noisy responses. We re-analyzed data from single neurons in the visual cortical motion area MT responding to dot motion in eight equally spaced directions [12, 13]. A classic model of these responses is a single cycle sinusoid as a function of the direction of dot motion with the free parameters phase, amplitude, and average firing rate. We chose this MT data set as the first example application because it has a high number of repeats ($n = 10$) and a low dimensional model, thus it is simple to visually inspect whether the neuronal tuning curves agree with the model predictions.

An example of an MT neuron direction tuning curve (Fig 7A, orange trace) has an excellent sinusoidal fit (blue trace), as reflected in its estimated $\hat{r}^2_{\mathrm{ER}} = 1.0$. Furthermore, the short confidence interval ([0.99, 1.0]) quantifies the lack of ambiguity about the quality of the fit. On the other hand, the tuning curve of a neuron with $\hat{r}^2_{\mathrm{ER}} = 0.05$ (Fig 7B) has a clear systematic deviation from the least-squares fit. Here the tuning curve is double-peaked and thus largely orthogonal to any single cycle sinusoid. It is important to notice that this neuron has far lower SNR (2.8 here vs. 20 for the example in A), as quantified by our estimator, $\widehat{\mathrm{SNR}}$ (Eq 16), which corrects for trial-to-trial variability (described below and defined in Methods, "Estimators of correction terms"). Thus without $r^2_{\mathrm{ER}}$, there would be plausible doubts about whether the correlation was lower because of noise or systematic deviation. Furthermore, with low SNR it would be plausible that the estimate itself is noisy (Fig 3A), but the short confidence interval ([0.01, 0.11]) unambiguously characterizes the fit as being systematically poor.

In some cases, neurons show little tuning for direction and thus have very low SNR over a set of directional stimuli. This in turn can cause $\hat{r}^2_{\mathrm{ER}}$ to give wild estimates (Fig 7C, $\widehat{\mathrm{SNR}} = 0.05$, $\hat{r}^2_{\mathrm{ER}} = 1.81$). If we truncate the value to the nearest possible $r^2_{\mathrm{ER}} = 1$, we might be

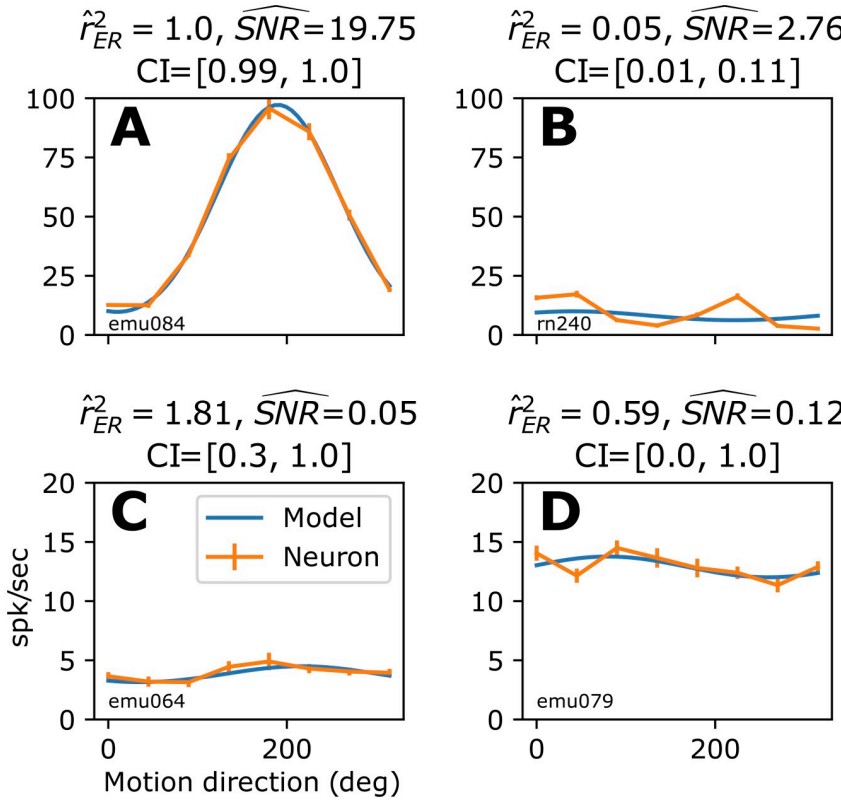

**Fig 7. Applying our unbiased estimator with CIs to fit four example MT neuronal direction tuning curves to a sinusoidal model. (A)** Example neuron tuning curve (orange trace with SEM bars) with excellent fit to sinusoidal model (blue trace, $\hat{r}^2_{ER} = 1.0$), high SNR and tight CI (parameters specified above plot panel). **(B)** Example neuron with poor fit to sinusoidal model but with a reasonable SNR and narrow CI that provide confidence that the neuronal tuning systematically deviates from the model. **(C)** Example neuron with poor SNR and wild estimate of $\hat{r}^2_{ER}$, which is reflected in large CI = [0.3, 1], suggesting that no conclusion can be made about how well the model describes any actual tuning here. **(D)** Example neuron with a seemingly reasonable $\hat{r}^2_{ER}$, but the low SNR and CI covering the entire interval [0, 1] reveals that this fit cannot be trusted.

tempted to interpret this as a well-fit direction selective neuron. But, the CI covers most of the interval of possible values ([0.3, 1]), making it clear that little information can be gleaned about $r^2_{ER}$ from this data. Extreme $\hat{r}^2_{ER}$ values themselves can indicate when the estimator is unreliable, but even a reasonable seeming $\hat{r}^2_{ER}$ value, for example $\hat{r}^2_{ER} = 0.59$ (Fig 7D), can be unreliable when there is a low $\widehat{SNR}$ (0.12). In this case, the confidence interval covers the maximal range ([0, 1]), indicating that the point estimate is unreliable. Thus, $\hat{r}^2_{ER}$ and its associated confidence interval quickly and unambiguously show how well the model fits the MT data, avoiding the tiresome and unreliable process of judging each fit by eye for the 162 neurons.

While we have shown to a good approximation that $\hat{r}^2_{ER}$ is unbiased and its expected value is largely invariant to SNR, this is definitely not the case for the variance of the estimator. Fig 3A shows clearly that the variability of the estimator is larger for lower SNR. This fact should be kept in mind when interpreting the spread of $\hat{r}^2_{ER}$ values. For example, we calculated $\hat{r}^2_{ER}$ and confidence intervals across our entire population of MT neurons. Of the estimates with high SNR (Fig 8, right side, $\widehat{SNR} > 3.5$), most neurons are well fit to the model and only a few have less than 3/4 of their variance accounted for (8/81). For the estimates with low SNR ($\widehat{SNR} < 3.5$), left side of Fig 8), this fraction is substantially higher (39/81), but the increased

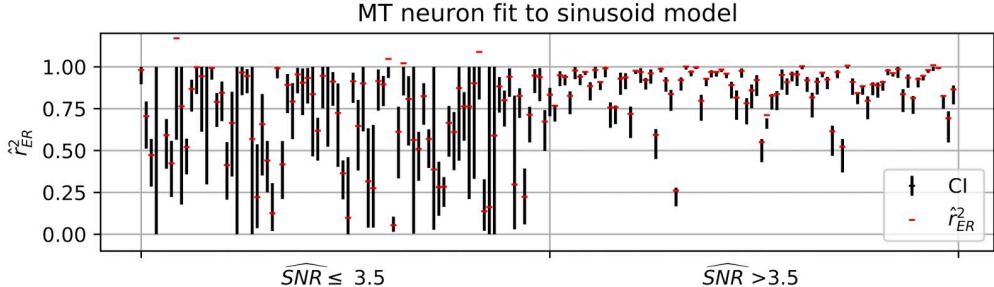

**Fig 8. Confidence intervals ($\alpha$ = 90%, vertical lines) and point estimates (red dashes) for $\hat{r}^2_{ER}$ across all MT neuron direction tuning curves fit to sinusoidal model.** Data points are grouped into two intervals on the basis of $\widehat{SNR}$ (of the direction tuning curves) being less than or equal to or higher than the median value (3.5), revealing that lower SNR (left interval) is associated with much longer CIs.

variability of these estimates will spread out the distribution, thus this difference in quantiles must be interpreted carefully. When estimating population dispersion, conclusions may be confounded by the SNR-dependence of the variability of $\hat{r}^2_{ER}$.

Comparing the naive $\hat{r}^2$ to our unbiased $\hat{r}^2_{ER}$ (Fig 9), the high SNR units (red points), lie close to the diagonal. Thus for these units, one could exchange the two estimates and come to similar general conclusions about model fits. The utility of $\hat{r}^2_{ER}$ is that it removes ambiguity about whether trial-to-trial variability may be spuriously pushing fits down (black points). The interpretation of the naive estimator $\hat{r}^2$ remains ambiguous for any given unit until it can be confirmed it does not suffer from this issue.

The MT data considered here has relatively few stimuli and many repeats, but other experimental paradigms involve a larger number of stimuli and, consequently, fewer repeats. Below we apply $\hat{r}^2_{ER}$ in these more challenging conditions.

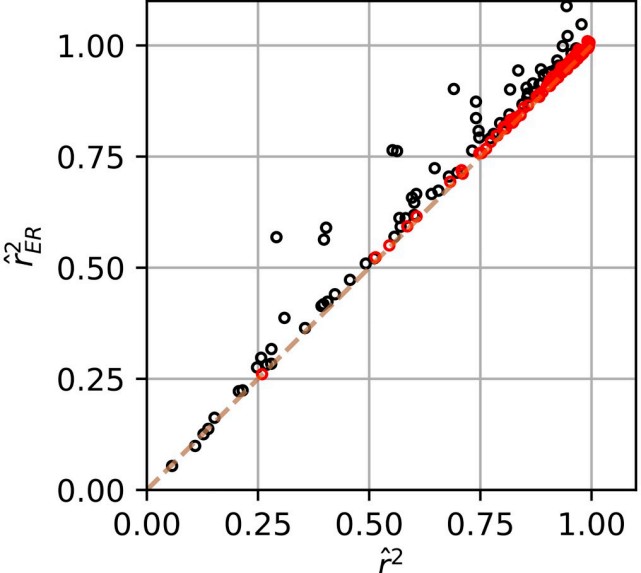

**Fig 9. Relationship of naive $\hat{r}^2$ and corrected $\hat{r}^2_{ER}$ between fits of sinusoidal model to MT data.** Units with $\widehat{SNR}$ greater than the median across the population ($\widehat{SNR} = 3.5$) are plotted in red and those less than or equal to in black.

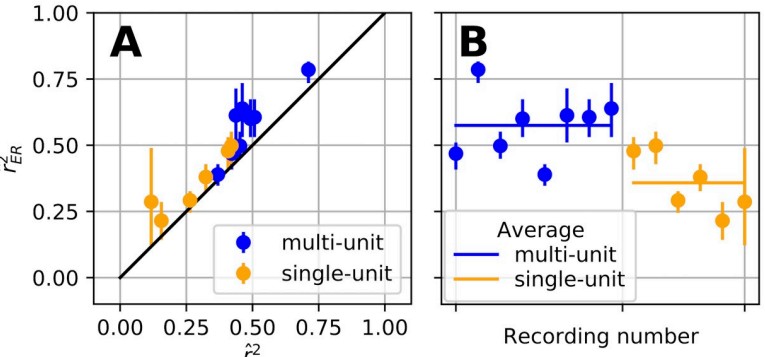

**Fig 10. Applying $\hat{r}^2_{\text{ER}}$ to analyze performance of a deep neural network (DNN) in predicting V4 responses to natural images. (A)** For single-unit (orange) and multi-unit (blue) recordings, $\hat{r}^2_{\text{ER}}$ is plotted against the naive $\hat{r}^2$. The relatively short $\alpha$ = 0.1 CIs (vertical bars) suggest that most of these correlation values are trustworthy. **(B)** The mean $\hat{r}^2_{\text{ER}}$ value across multi-unit recordings (horizontal blue line) is significantly higher than that for the set of single-unit recordings (orange horizontal line; Welch's t-test t = 3.7, p = 0.005). Because individual estimates are asymptotically unbiased, the group average inherits this lack of bias.

## Application of estimator to V4 data

The primate mid-level visual cortical area V4 is known to have complex, high-dimensional selectivity for visual inputs. To rigorously assess models of neuronal responses in areas like V4, validation needs to be performed on responses to a large corpus of natural images to ensure that models capture ecologically valid selectivity [14, 15]. Thus, the number of unique stimuli, $m$, will be large at the expense of having relatively few repeats, $n$, and SNR can be low because stimuli are not customized to the preferences of a given neuron. These are the challenging conditions under which $\hat{r}^2_{\text{ER}}$ avoids the major confounds of $\hat{r}^2$. Here we estimate $\hat{r}^2_{\text{ER}}$ and associated 90% confidence intervals for a model that won the University of Washington V4 Neural Data Challenge by most accurately predicting single-unit (SUA) and multi-unit activity (MUA) for held-out stimuli (see Methods, "Electrophysiological data"). Plotting $\hat{r}^2_{\text{ER}}$ against $\hat{r}^2$ (Fig 10A) shows that the corrected estimates are higher than the naive estimates (points lie above diagonal line). Using $\hat{r}^2_{\text{ER}}$ here is important because it provides confidence that the poor fit quality is not a result of noise and that the best performing model often did not explain more than 50% of the variance in the tuning curve.

While we have examined $\hat{r}^2_{\text{ER}}$ for individual recordings, it can also be useful to estimate the average quality of model fit across a population of neurons. Since the individual estimates are unbiased, the group average is also an unbiased estimate of the population mean $\hat{r}^2_{\text{ER}}$. We computed such group means for the single-unit and multi-unit V4 recordings (Fig 10B), and found that the model performed significantly better in predicting the responses of multi-unit activity (Welch's t-test p = 0.005, MUA mean = 0.57, SUA mean = 0.35). If instead the naive $\hat{r}^2$ were used, this finding could have been dismissed as the result of MUA having higher SNR and thus naturally higher $\hat{r}^2$. As it stands, this interesting observation can be followed up to potentially gain insight about the structure of selectivity across multiple units recorded nearby in V4.

Finally, this V4 data set provides a good example of how using $\hat{r}^2_{\text{ER}}$ can allow testing a larger stimulus space, as predicted by simulations above in Fig 3B. Fig 11 shows that with $\hat{r}^2_{\text{ER}}$ (solid lines, on average two trials is enough to estimate the true correlation, whereas the naive estimator requires more repeats (higher $n$) to converge. For example, for recording 1 (red), $\hat{r}^2_{\text{ER}}$ (solid

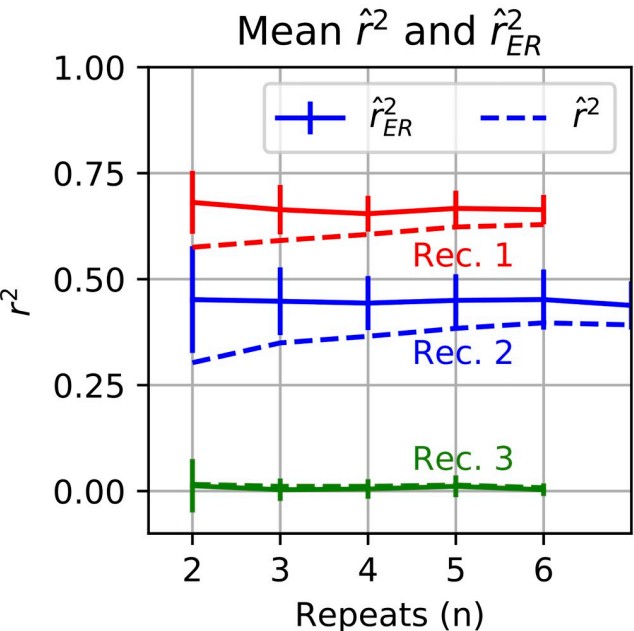

**Fig 11. Relationship of naive $\hat{r}^2$ and corrected $\hat{r}^2_{ER}$ with *n*, the number of repeats for V4 data.** Different colors indicate different recordings. Solid lines show the average $\hat{r}^2_{ER}$ estimate across random shuffling of trials (with replacement); vertical bars indicate SD. Dashed lines show average $\hat{r}^2$.

line) on average predicts the same quality of model fit for two or more stimulus repeats, whereas even after six repeats, the naive $\hat{r}^2$ has not converged.

## Signal-to-noise ratio as recording quality metric

We have shown above that correcting for bias in $r^2$ is important, but it is also critical to recognize when recordings are so noisy that they are effectively useless for evaluating a model. Here we demonstrate the use of the signal-to-noise ratio (SNR) as a quality metric to help make this determination. We define the SNR for a neuronal tuning curve to be the ratio of the variation in the expected response across stimuli to the trial-to-trial variability across repeats:

$$\text{SNR} = \frac{\frac{1}{m}\sum_{i=1}^{m}\left(\mu_i - \bar{\mu}\right)^2}{\sigma^2}, \tag{5}$$

where $\mu_i$ is the expected response to the *i*th stimulus and $\bar{\mu} = \frac{1}{m}\sum_{i=1}^{m}\mu_i$. For experimental data, we do not know $\mu_i$ in Eq 5, and rather than substituting sample estimates, $Y_i$, which would give an inflated estimate, we use an equation that corrects for trial-to-trial noise ($\widehat{\text{SNR}}$, Eq 16, Methods). The removal of this bias in our SNR metric allows for direct comparisons between studies with different numbers of repeats and amounts of trial-to-trial variability. This is appropriate because SNR is not a function of *n* or *m*, rather it can vary across neurons, sets of stimuli and recording modalities, as we show below.

We examined a diverse collection of neural data sets (see Methods, Electrophysiological data) and found wide variation in $\widehat{\text{SNR}}$ both within and across the data sets (Fig 12A). At the low end, calcium imaging data from cortical neurons in area VISp of mouse responding to gratings (pink trace $N$ = 40,520 neuronal ROIs, [16]) had a median SNR of 0.01, while at the high end, MT neurons in response to dot motion [12] had a median $\widehat{\text{SNR}}$ of 3.5 (blue trace,

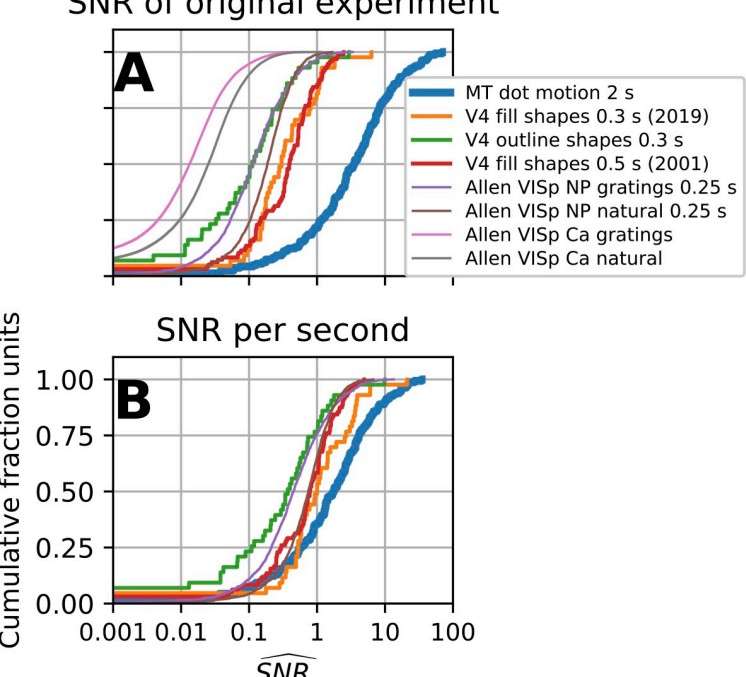

**Fig 12. A comparison of our data quality metric, the signal-to-noise ratio estimator $\widehat{SNR}$ (Eq 16), across several datasets. (A)** The cumulative distribution of $\widehat{SNR}$ under the original experimental protocols. Traces with the same line thickness have similar numbers of $n$ and $m$. Thick line (blue): MT data has $n \approx 10$, $m = 8$. Medium lines (green, orange, red): V4 data has $n \approx 5$, $m \approx 350$. Thin lines: Allen Inst. data has $n \approx 50$, $m \approx 120$. The Allen Inst. data has two recording modalities: extracellular action potentials (spikes) on Neuropixel probes (NP) and two-photon calcium imaging (Ca). Both were recorded for the same stimuli: natural scenes and gratings (see Methods, "Electrophysiological data"). **(B)** Distribution of $\widehat{SNR}$ after normalization with respect to the duration of the spike counting window (traces for calcium signal are not included). The normalization assumes that the original average spike rate can be applied to a 1 s counting window. But, if firing rates tend to decay over time, this will produce overestimates for recordings shorter than 1 s and underestimates for recordings longer than 1 s.

$N$ = 162). A stimulus protocol nearly identical to that used for the VISp Ca$^{2+}$ data (pink and gray traces for gratings and natural images, respectively) was used to collect the Allen Institute NeuroPixel electrode data [17] (purple and brown traces $N$ = 2,015); however, the Ca$^{2+}$ data had a substantially lower $\widehat{SNR}$ (0.01 and 0.02) compared to the electrode data ($\widehat{SNR}$ 0.12 and 0.19), suggesting that this difference relates to the recording modality.

In the case of spiking neurons, SNR can be improved by increasing the stimulus duration and thus the spike counting window. Under the generally optimistic assumption that spike rate stays constant in the counting window and assuming that the spike counting window could be changed given experimental constraints, we can normalize $\widehat{SNR}$ across the data-sets to what the $\widehat{SNR}$ would have been had all spike count windows been 1 second long (Fig 12B). Under these assumptions, this normalization allows us to examine SNR differences across studies removing the counting window length as a factor. We find this reduces the differences in $\widehat{SNR}$ across the spiking data-sets (the six right-most traces), thus the outstanding $\widehat{SNR}$ of the MT data-set could potentially have been achieved if spike count windows had been longer for the other experiments. Still, of the spiking data, the Allen Neuropixel data has the lowest medians, thus additional efforts to ameliorate low SNR (via number of trials or stimulus choice) could be utilized. Furthermore, the assumption of a constant spike rate will hold to

different degrees: neural responses can peak shortly after stimulus onset and then return close to baseline. Thus, different experimental conditions call for different standards for number of trials and stimulus duration to adequately characterize a tuning curve.

To provide concrete meaning to $\widehat{SNR}$, we suggest interpreting it in terms of the number of trials ($m$ and $n$) needed to reliably detect stimulus modulation in an $F$-test. Specifically, for a given $m$ and $n$ we computed the minimal SNR required to achieve a high probability ($\beta = 0.99$) of rejecting the null hypothesis that the mean response to all stimuli is the same (see Methods, SNR relation to $F$-test and number of trials, Eq 22). We plot a color map of this minimal SNR as a function of $m$ and $n$ (Fig 13), where the diagonal grey contour lines indicate fixed total number of trials ($mn$) for different $m$: $n$ ratios. In general, as the total number of trials increases (moving perpendicular to the grey diagonals toward the upper right), the SNR required for reliable tuning curve estimates decreases. The SNR threshold is lower when $n$ is favored over $m$ for the same number of total trials, i.e., the SNR threshold level iso-contours have steeper slopes than the grey diagonals.

On this map, we can locate points corresponding to the $m$ and $n$, roughly, for data sets in Fig 12. The three V4 data sets have about the same number of stimuli and repeats (arrow marked "V4", Fig 13), and thus require SNR≈0.1 or greater, implying that from 3% to 23% of the V4 data does not pass the criterion (Fig 12A, red and green traces, respectively, define endpoints). The MT data has the fewest number of total trials and thus has the highest threshold SNR ≈ 0.5, which leaves 10% of the neurons with poorly estimated tuning curves. If more stimuli had been used at the expense of fewer repeats, say $n = 2$ and $m = 40$, then only a quarter of the neurons would have exceeded the increased threshold of SNR > 1. The Allen Ca$^{2+}$ and spike data sets both had similar $m$ and $n$. Relative to the other data sets they had far more total trials and a greater number of repeats, thus the SNR criterion is substantially lower (SNR > 0.01). Still, for the Ca$^{2+}$ data, ∼ 37% of the grating and ∼ 25% of the natural image data did not have reliable tuning (Fig 12A, pink and grey thin trace). The Allen spiking data on the other hand had much higher SNR, and thus more trials could have been spent on expanding the stimulus set and fewer on repeated presentation (Fig 12A, thin brown purple trace).

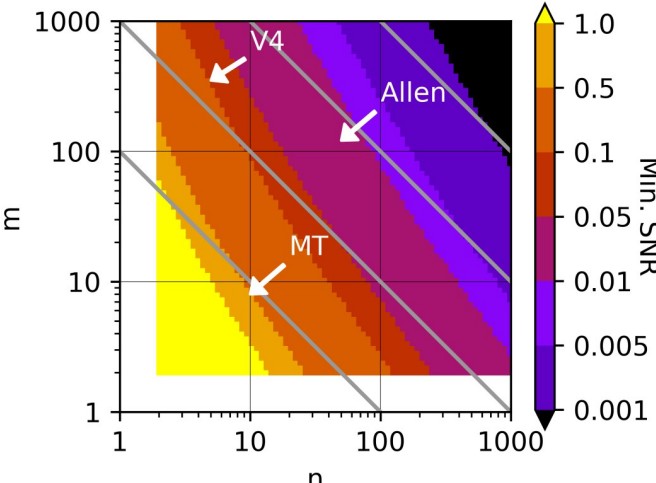

**Fig 13. The minimal SNR needed to reliably detect tuning as a function of $m$, the number of unique stimuli, and $n$, the number of repeats of each stimulus.** White arrows indicate the approximate location in ($m$, $n$) corresponding to the datasets used in Fig 12. Gray diagonal lines indicate constant number of total trials ($n \times m$).

We have shown SNR can be employed as a simple metric with a concrete interpretation to judge data quality across different organisms, recording modalities and brain regions for the purpose of making comparative analyses and setting aside data that has little or no power. The expected distribution of SNR, based on prior data, can be taken into account when choosing $n$ and $m$ to achieve a criterion level of statistical power for an experiment. If SNR is high, recording time can be reduced by keeping $n$ and $m$ low, or a larger stimulus set can be explored by increasing $m$ at the expense of $n$.

## Discussion

### Summary

We have investigated the estimation of the correlation between a model prediction and the expected response of a neuron. Although it has been long established that trial-to-trial variability will cause the classic estimator, Pearson's $\hat{r}^2$, to underestimate correlation, there has been no direct comparison of prior methods to account for this confound. We found that some methods grossly over estimate correlation in high noise conditions, and we built upon the best performing method to derive a more generally applicable estimator, $\hat{r}^2_{ER}$, that performs as well as or better than prior methods. We analytically validated $\hat{r}^2_{ER}$ by determining that it was a consistent estimator in the number of stimuli. We found in simulation that it had a small upward bias, but this was only appreciable at very high noise levels. None of the prior methods that we examined had validated confidence intervals, thus we developed confidence intervals for $\hat{r}^2_{ER}$. Motivated by the failure of generic bootstrap methods to achieve satisfactory confidence intervals, we developed a confidence interval method that outperformed them.

Applying our estimator to neural data, we demonstrated its essential value. In the case of MT recordings, it was able to unambiguously distinguish neurons for which a sinusoidal model was a good fit from those for which it was a poor fit specifically because of systematic deviation and not because of noise. The associated confidence intervals allowed the systematic identification of noisy recordings that served no practical use in assessing the fit of the model. Poor model fits caused by noise vs. those caused by systematic differences in selectivity have very different interpretations, yet the traditional $\hat{r}^2$ does not differentiate them while $\hat{r}^2_{ER}$ does.

Application of the estimator to the winning UW neural data challenge model, a deep neural network (DNN), provides the only validated assessment of state-of-the-art predictive model performance in V4. The estimator along with its CIs identified neurons that were challenging to the DNN and perhaps require a different modeling approach. It also validated the existence of single units that had nearly 50% of their variance explained, indicating that the DNN functionally captured a substantial part of what these units encode across natural images and thus could provide real insight into naturalistic V4 single unit encoding. On a practical level, we showed how the estimator allows for gains in trial efficiency since it converges more rapidly than $\hat{r}^2$ (Figs 3B and 11). This is important when many stimuli are needed to validate models of high dimensional neural tuning.

Our tests on experimental data revealed that some neurons had confidence intervals covering the entire range of possible values, motivating us to propose the signal-to-noise ratio (SNR) as a metric of neural recording quality in the context of model fitting. We provide an unbiased estimator of SNR (Eq 16) and a practical interpretation: for a given number of stimuli and repeats, the SNR should be sufficient to reliably detect stimulus-driven response modulation on the basis of an F-test (Eq 22). Examining a variety of data sets, we found differences with respect to how the numbers of stimuli vs. repeats ($m$ vs. $n$) were balanced, revealing how adjustments can be made on the basis of SNR to improve experimental efficiency. We also found large differences in SNR across data sets that are likely related to recording modality

## Interpretation of $r^2_{\text{ER}}$

We have introduced an estimator and confidence intervals for the correlation between the true tuning curve of a neuron (its expected value across stimuli) and model predictions: $\hat{r}^2_{\text{ER}}$. In the context of sensory neurophysiology, we believe it is reasonable to think of $r^2_{\text{ER}}$ as reflecting solely how well a model explains a sensory representation. We justify this by the fact that $r^2_{\text{ER}}$ is solely a function of E[Response|Stimulus] thus solely a function of stimuli. We note two caveats: (1) non-sensory signals can influence sensory responses, e.g., eye movements which may be stimulus dependent and (2) E[Response|Stimulus] is not the only component of the sensory response, e.g., variability can also be stimulus dependent [18].

## Relationship of $\hat{r}^2_{\text{ER}}$ to $\Upsilon$

We have taken a similar approach to Haefner and Cumming [6], commensurately their estimator gives nearly identical results to ours in simulation (less than $< 0.0001\%$ power unexplained for SNR = 1 vs $>1\%$ for the other estimators), though we provide an important generalization. Their formula requires the calculation of the sample variance because their derivation relies on the F-distribution formed by taking the ratio of the sum of squares of model residual over the sample variance (see Methods, $\Upsilon$). This is problematic if stimuli are never repeated in an experiment (for example, in free viewing experiments), then one has to assume a priori the trial-to-trial variability either from previous experimental measurements or by asserting a theoretical mean-variance relationship (e.g., the square root of Poisson distributed spiking gives $\sigma^2 = \frac{1}{4}$).

Haefner and Cumming's estimator is more general than the $\hat{r}^2_{\text{ER}}$ we have presented. The estimate $\hat{r}^2_{\text{ER}}$ measures the variance of the mean centered data explained by a single covariate, and $\Upsilon$ measures the variance explained by a linear combination of up to $m$ covariates. In particular, $\Upsilon$, accounts for the decrease in degrees of freedom available to the noise as more covariates are added. We also provide the more general version of our estimator for the case of variance explained by a linear model (Eq 23) with the advantage discussed in the previous paragraph.

## SNR

We found that differences in SNR can be substantial and widely varying across neurons, data sets, and recording methods. Given the rise in large scale recordings and sharing of neuronal data, we believe unbiased estimates of SNR should be reported so that researchers can quickly judge whether a data set has sufficient statistical power or whether its power is on par with that of data sets from potentially comparable studies. We provide concrete criteria by which to interpret SNR: the statistical power to detect stimulus-driven response modulation. Strikingly, in our small sample of data sets, many neurons do not pass this criteria, suggesting that the adoption of a standard criterion for data quality, such as our SNR metric, could have a major impact in practice. Furthermore, guided by the metric the experimentalist can take steps to improve SNR by increasing stimulus duration and associated spike counting windows or by customizing stimuli to the preferences of a neuron. On the other hand, the deleterious effects of low SNR can be ameliorated by favoring repeats over number of stimuli (Fig 13).

One conceptual interpretation of the SNR metric we introduced is that it quantifies, for the time scale of the spike count window, the overall variance in the responses of the neuron attributable to the tuning curve of the neuron vs. trial-to-trial variability about that tuning curve. For example on the time scale of 1 second, a large fraction of spike-based recordings had SNR>1, indicating that more variance was caused by the stimulus than by other sources (Fig 12B blue, orange, green traces median SNR > 1). Still, an appreciable number of neurons were dominated by their trial-to-trial variability. Whether this is the result of stimulus choice and perhaps would be different in a more natural context is an open question. Recent theoretical and experimental work has argued that weakly tuned and untuned neurons can contribute to sensory encoding [19–21]. The corrected estimate of SNR we provide (Eq 16) along with naturalistic stimulation can help to identify such neurons.

## Further work

Small improvements to our $\hat{r}^2_{\mathrm{ER}}$ estimator could be made by decreasing its bias in the case of very low SNR (see Methods, "Bias of $\hat{r}^2_{\mathrm{ER}}$"). In the case of very low SNR, a single neuronal recording has little inferential power, but across a population of neurons, estimates of the average correlation to a model's predictions can have low enough variance to provide useful inference. Yet, at very low SNR an appreciable bias begins to appear that will remain in the population average. We showed this bias is a function of the covariance between the numerator ($\hat{C}_{\mathrm{ER}_m}$) and inverse of the denominator ($\frac{1}{\hat{V}_{\mathrm{ER}_m}}$) of $\hat{r}^2_{\mathrm{ER}}$ and Jensen's gap where $\mathrm{E}[\frac{1}{\hat{V}_{\mathrm{ER}_m}}] > \frac{1}{\mathrm{E}[\hat{V}_{\mathrm{ER}_m}]}$ (Eq 17). The former covariance can be removed by using separate subsets of the data for estimation of the numerator and denominator. To reduce the influence of Jensen's gap, further work could attempt to directly estimate and correct for its value. In addition, analytic results on how this small sample bias varies as a function of critical parameters $m$, $n$ and SNR would be helpful in it's interpretation.

In the derivation of our estimator we assume the $m$ responses across which the model predictions are evaluated are independent. Thus the estimator in its current form would not be appropriate for evaluating models that make predictions across spike counts in adjacent time bins. In future work we plan to extend our estimator to the case of correlated responses.

Here we have derived an estimator for the case where deterministic model predictions are correlated to a noisy signal. Often, one noisy signal is correlated to another, for example when judging the similarity of tuning curves from two neurons (termed signal correlation). We have extended the methods described here to the neuron-to-neuron case and will describe this in a forthcoming publication.

A subtle but important point about our estimator is that it assumes stimuli are fixed: it estimates the $r^2_{\mathrm{ER}}$ for the exact stimuli shown. An investigator may be interested in the quality of a model across a large corpus of natural images of which only a small fraction can be included in a recording session. In this case, one collects responses for a random sample of images, fits the model to some (training set) and tests the model on others (test set). The random test set will account for over-fitting and using $\hat{r}^2_{\mathrm{ER}}$ will account for noise in the neural responses in the evaluation on the test set. Crucially though, this does not account for the variability in the parameters of the model induced by the random training sample. Intuitively, estimated model parameters will vary across image sets even in the absence of trial-to-trial variability. The correct interpretation of $\hat{r}^2_{\mathrm{ER}}$ in this case is that it estimates how well a model can perform given finite noisy training data on noiseless test set data, and *not* as the best the model could possibly perform given infinite training data. Indeed, with more neural responses and less noise, model test set performance would improve. David and Gallant [5] explored this issue calling it 'estimation noise' and provided an extrapolation method for estimating the fit of a linear model

given unlimited stimuli. The estimator was not evaluated in terms of its bias or variance, and no analytic solutions that directly remove the bias of finite training data have been proposed. Both are valuable directions to pursue: the former to build confidence in the current method and the latter for potential gains in trial efficiency. A data driven re-sampling approach may be unavoidable when evaluating complex models where the relationship between the amount of training samples and model performance would be analytically intractable, such as a deep neural network or biophysical model.

## Materials and methods

### Simulation procedure

To simulate model-to-neuron fits, the square root of neural responses, $r_{i,j}$, for the $i$th of $m$ stimuli and the $j$th of n trials are modeled as independent normally distributed responses:

$$Y_{i,j} \sim N[\mu_i, \sigma^2], \tag{6}$$

where variance $\sigma^2$ is the same across all $Y_{i,j}$. The mean response of the neuron to the $i$th stimulus (tuning curve) is $\mu_i = a + b \sin\left(\frac{(i-1)2\pi}{m} + \theta\right)$ (Fig 1A, green trace solid dots) whose correlation to the model predictions $\mu_i = \sin\left(\frac{(i-1)2\pi}{m}\right)$ (red trace solid dots) are estimated, and the true correlation is $r_{ER}^2 = \cos^2(\theta)$. The results of the simulation are only a function of the magnitude of the centered vector of expected responses $d^2$ the correlation between model prediction and tuning curves, $m$, $n$, and $\sigma^2$ thus the form of the model and true tuning curve is arbitrary. We choose a sinusoid for the simplicity of adjusting the phase, $\theta$, to simulate different $r_{ER}^2$.

From this model we draw $n$ responses for each of the $m$ stimuli and apply our estimator to this sample. We repeat this across many IID simulations to accumulate reliable statistics.

### Assumptions and terminology for derivation of unbiased estimators

Below we derive an unbiased estimator of the fraction of variance explained when a known signal is being fit to noisy neural responses. For this derivation, we assume the responses have undergone a variance stabilizing transform such that trial-to-trial variability is the same across all stimuli. For example, if the neural responses are Poisson distributed, $Y_{i,j} \sim P(\lambda_i)$, where $Y_{i,j}$ is the response to the $j$th repeat of the $i$th stimulus, which has expected response $\lambda_i$, then a variance stabilizing transform is the square root. In particular, if $Y_{i,j}^* = \sqrt{Y_{i,j}}$, then,

$$E[Y_{i,j}^*] = E[\sqrt{P(\lambda_i)}] \approx \sqrt{\lambda_i},$$

and

$$\text{Var}[Y_{i,j}^*] = \text{Var}[\sqrt{P(\lambda_i)}] \approx \frac{1}{4}.$$

The expected value of the transformed response, $Y_{i,j}^*$, still increases with $\lambda_i$, whereas the variance is now approximately constant. To improve the estimate of the mean response, $n$ repeats of each stimulus are collected. Invoking the central limit theorem, we can make the approximation:

$$\frac{1}{n}\sum_{j=1}^{n} Y_{i,j}^* = \bar{Y}_i^* \sim N\left(\sqrt{\lambda_i}, \ \frac{1}{4n}\right),$$

where the average across the $n$ repeats is approximately normally distributed with variance decreasing with $n$. The assumption of a Poisson distributed neural response is not always

accurate. A more general mean-to-variance relationship,

$$\sigma^2(\mu) = a\mu^b,$$

can be approximately stabilized to 1 by,

$$f(x) = [\sqrt{a}(1 - \frac{1}{2}b)]^{-1} x^{1-\frac{1}{2}b}.$$

A square root will stabilize any linear mean-to-variance relationship ($b = 1$), but an unknown slope, $a$, requires that this parameter be estimated. In the case of the linear relationship, this simply requires taking a square root and then averaging the estimated variance, which is constant, across all stimuli. To account for more diverse mean-to-variance relationships, the Box-Cox technique can be used find an appropriate exponent by which to transform the data [22]. For the derivation below, we assume that variance-stabilized responses to $n$ repeats have been averaged for each of $m$ stimuli to yield the mean response to the $i$th stimulus: $Y_i \sim N\left(\mu_i, \frac{\sigma^2}{n}\right)$, where $\sigma^2$ is the trial-to-trial variability and $\mu_i$ the $i$th expected value.

## Unbiased estimation of $r^2$

Given a set of mean neural responses, $Y_i$, and model predictions, $v_i$, the naive estimator, $\hat{r}^2$, is calculated as follows:

$$\hat{r}^2 = \frac{\left(\sum_{i=1}^m (v_i - \bar{v})(Y_i - \bar{Y})\right)^2}{\sum_{i=1}^m (v_i - \bar{v})^2 \sum_{i=1}^m (Y_i - \bar{Y})^2}. \tag{7}$$

Our goal is to find an estimator such that,

$$\mathrm{E}[\hat{r}^2_{\mathrm{ER}}] = r^2_{\mathrm{ER}} = \frac{\left(\sum_{i=1}^m (v_i - \bar{v})(\mu_i - \bar{\mu})\right)^2}{\sum_{i=1}^m (v_i - \bar{v})^2 \sum_{i=1}^m (\mu_i - \bar{\mu})^2}, \tag{8}$$

where $r^2_{\mathrm{ER}}$ is the correlation in the absence of noise, i.e., the fraction of variance explained by the model prediction, $v$, of the expected response (ER), $\mu_i$, of the neuron. Our strategy will be to remove the bias in the numerator and denominator separately and then reform the ratio of these unbiased estimators for an approximately unbiased estimator.

**Unbiased estimate of numerator.** The numerator of Eq 7, which we call $\hat{C}_m$, is a weighted sum of normal random variables that is then squared, thus it has a scaled non-central chi-squared distribution:

$$\hat{C}_m = \left(\sum_{i=1}^m (v_i - \bar{v})(Y_i - \bar{Y})\right)^2 \sim \frac{\sigma^2}{n} \sum_{i=1}^m (v_i - \bar{v})^2 \chi_1^2\left(\frac{\left(\sum_{i=1}^m (v_i - \bar{v})(\mu_i - \bar{\mu})\right)^2}{\frac{\sigma^2}{n} \sum_{i=1}^m (v_i - \bar{v})^2}\right), \tag{9}$$

and since $\mathrm{E}[\chi_m^2(\lambda)] = \lambda + m$ its expectation is:

$$\begin{aligned}
\mathrm{E}[\hat{C}_m] &= \frac{\sigma^2}{n} \sum_{i=1}^m (v_i - \bar{v})^2 \mathrm{E}\left[\chi_1^2\left(\frac{\left(\sum_{i=1}^m (v_i - \bar{v})(\mu_i - \bar{\mu})\right)^2}{\frac{\sigma^2}{n} \sum_{i=1}^m (v_i - \bar{v})^2}\right)\right] \\
&= \frac{\sigma^2}{n} \sum_{i=1}^m (v_i - \bar{v})^2 \left(\frac{\left(\sum_{i=1}^m (v_i - \bar{v})(\mu_i - \bar{\mu})\right)^2}{\frac{\sigma^2}{n} \sum_{i=1}^m (v_i - \bar{v})^2} + 1\right) \\
&= \left(\sum_{i=1}^m (v_i - \bar{v})(\mu_i - \bar{\mu})\right)^2 + \frac{\sigma^2}{n} \sum_{i=1}^m (v_i - \bar{v})^2.
\end{aligned} \tag{10}$$

In the final line, the term on the left is the desired numerator and the term on the right the bias contributed by $\sigma^2$. To form our estimator, $\hat{C}_{ER_m}$, for the numerator of Eq 15, we simply subtract an unbiased estimator of this bias term from the numerator of the naive estimator 2:

$$\hat{C}_{ER_m} = \left(\sum_{i=1}^{m}(v_i - \bar{v})(Y_i - \bar{Y})\right)^2 - \frac{\hat{\sigma}^2}{n}\sum_{i=1}^{m}(v_i - \bar{v})^2, \tag{11}$$

where $\hat{\sigma}^2$ is typically the sample variance, $s^2$, estimated from the data, but it can be any unbiased estimator, even an assumed constant. For example, if stimuli are not repeated (i.e., $n = 1$) and one is willing to assume that responses are Poisson distributed, then the square root of these responses will give $\sigma^2 = \frac{1}{4}$ and thus one can substitute $\hat{\sigma}^2 = \frac{1}{4}$. The case for the denominator is similar.

**Unbiased estimate of denominator.** The denominator of Eq 7, which we call $\hat{V}_m$, is a weighted sum of squared normal random variables and thus also follows a scaled non-central chi-squared distribution:

$$\sum_{i}^{m}(v_i - \bar{v})^2\sum_{i}^{m}(Y_i - \bar{Y})^2 \sim \frac{\sigma^2}{n}\sum_{i=1}^{m}(v_i - \bar{v})^2\chi_{m-1}^2\left(\frac{\sum_{i=1}^{m}(\mu_i - \bar{\mu})^2}{\frac{\sigma^2}{n}}\right), \tag{12}$$

with expectation,

$$\begin{aligned}E[\hat{V}_m] &= \frac{\sigma^2}{n}\sum_{i=1}^{m}(v_i - \bar{v})^2 E\left[\chi_{m-1}^2\left(\frac{\sum_{i=1}^{m}(\mu_i - \bar{\mu})^2}{\frac{\sigma^2}{n}}\right)\right] \\ &= \sum_{i=1}^{m}(v_i - \bar{v})^2\sum_{i=1}^{m}(\mu_i - \bar{\mu})^2 + (m-1)\frac{\sigma^2}{n}\sum_{i=1}^{m}(v_i - \bar{v})^2.\end{aligned} \tag{13}$$

Similarly to the numerator, the first term is the desired denominator, and the second term is the bias. Thus, we subtract an unbiased estimate of this second term from the naive denominator:

$$\hat{V}_{ER_m} = \sum_{i=1}^{m}(v_i - \bar{v})^2\sum_{i=1}^{m}(Y_i - \bar{Y})^2 - (m-1)\frac{\hat{\sigma}^2}{n}\sum_{i=1}^{m}(v_i - \bar{v})^2. \tag{14}$$

Taking the ratio of these two unbiased estimators (Eqs 11 and 14) we have:

$$\hat{r}_{ER}^2 = \frac{\hat{C}_{ER_m}}{\hat{V}_{ER_m}} = \frac{\left(\sum_{i}^{m}(v_i - \bar{v})(Y_i - \bar{Y})\right)^2 - \frac{\hat{\sigma}^2}{n}\sum_{i=1}^{m}(v_i - \bar{v})^2}{\sum_{i=1}^{m}(v_i - \bar{v})^2\sum_{i=1}^{m}(Y_i - \bar{Y})^2 - \frac{\hat{\sigma}^2}{n}\sum_{i=1}^{m}(v_i - \bar{v})^2(m-1)}. \tag{15}$$

This equation can be further simplified by scaling the model predictions such that $\sum_{i=1}^{m}(v_i - \bar{v})^2 = 1$.

**Estimators of correction terms.** Two important parameters, $d^2 = \frac{1}{m}\sum_{i=1}^{m}(\mu_i - \bar{\mu})^2$ and $\sigma^2$, are unknown. Below we provide unbiased estimators of each of these terms. An unbiased estimate of sample variance for trials of the $i$th stimulus is $s_i^2 = \frac{1}{n-1}\sum_{j=1}^{n}(Y_{i,j} - \bar{Y}_{i,\cdot})^2$, where the dot in the subscript of $\bar{Y}_{i,\cdot}$ indicates the mean over repeats. Assuming the variance is the same across stimuli, we can average over $i$ for a global estimate:

$$s^2 = \frac{1}{m}\sum_{i=1}^{m}s_i^2.$$

Throughout the paper we use this as our estimate of trial-to-trial variability $\hat{\sigma}^2$.

For $d^2$ we have:

$$\mathrm{E}[\frac{1}{m}\sum_{i=1}^{m}(Y_i - \bar{Y})^2] \quad = \frac{1}{m}\mathrm{E}\left[\frac{\sigma^2}{n}\chi_{m-1}^2\left(\frac{n}{\sigma^2}\sum_{i}^{m}(\mu_i - \bar{\mu})^2\right)\right]$$

$$= \frac{1}{m}\left(\sum_{i}^{m}(\mu_i - \bar{\mu})^2 + (m-1)\frac{\sigma^2}{n}\right),$$

which would be inflated by trial-to-trial variability, so as an unbiased estimator we use,

$$\hat{d}_{\mathrm{ER}}^2 = \frac{1}{m}\left(\sum_{i=1}^{m}(Y_i - \bar{Y})^2 - (m-1)\frac{\hat{\sigma}^2}{n}\right).$$

We use this estimator to correct the estimate of SNR (Eq 5) for trial-to-trial variability as follows:

$$\widehat{\mathrm{SNR}} = \frac{\hat{d}_{\mathrm{ER}}^2}{\hat{\sigma}^2}. \tag{16}$$

## Bias of $\hat{r}_{\mathrm{ER}}^2$

To remove the bias of Pearson's $\hat{r}^2$, we follow the approach of subtracting off its effect in the numerator and denominator. Prior work has not examined the potential problem with this approach: the expectation of a non-linear transformation of a set of random variables is not necessarily the transformation of their expected values. In this particular case, the expectation of the ratio is not necessarily the ratio of the expectations: $\mathrm{E}[\hat{C}_{\mathrm{ER}_m}/\hat{V}_{\mathrm{ER}_m}] \neq \mathrm{E}[\hat{C}_{\mathrm{ER}_m}]/\mathrm{E}[\hat{V}_{\mathrm{ER}_m}]$. Thus even though we have removed the bias in the numerator and denominator, it does not imply their ratio is unbiased. Calculating the expectation of the ratio we see the conditions under which it will be unbiased:

$$\mathrm{E}\left[\hat{C}_{\mathrm{ER}_m}/\hat{V}_{\mathrm{ER}_m}\right] = \mathrm{E}\left[\hat{C}_{\mathrm{ER}_m}\frac{1}{\hat{V}_{\mathrm{ER}_m}}\right] = \mathrm{Cov}\left[\hat{C}_{\mathrm{ER}_m}, \frac{1}{\hat{V}_{\mathrm{ER}_m}}\right] + \mathrm{E}\left[\hat{C}_{\mathrm{ER}_m}\right]\mathrm{E}\left[\frac{1}{\hat{V}_{\mathrm{ER}_m}}\right]. \tag{17}$$

Thus, $\hat{r}_{\mathrm{ER}}^2$ is unbiased if $\mathrm{Cov}\left[\hat{C}_{\mathrm{ER}_m}, \frac{1}{\hat{V}_{\mathrm{ER}_m}}\right] = 0$ and $\mathrm{E}\left[\frac{1}{\hat{V}_{\mathrm{ER}_m}}\right] = \frac{1}{\mathrm{E}[\hat{V}_{\mathrm{ER}_m}]}$, but we find in simulation often $\mathrm{Cov}\left[\hat{C}_{\mathrm{ER}_m}, \frac{1}{\hat{V}_{\mathrm{ER}_m}}\right] \neq 0$ and by Jensen's inequality $\mathrm{E}\left[\frac{1}{\hat{V}_{\mathrm{ER}_m}}\right] \geq \frac{1}{\mathrm{E}[\hat{V}_{\mathrm{ER}_m}]}$.

Thus if the estimator $\hat{r}_{\mathrm{ER}}^2$ is not unbiased for $r_{\mathrm{ER}}^2$ what recommends it over the naive $\hat{r}^2$? While we mainly focused on how in simulation for typical ranges of parameters it has a lower bias (Fig 3) it also has a theoretical justification. As we saw in simulation, as the number of stimuli, $m$, increases, its bias diminishes while that of $\hat{r}^2$ does not (Fig 3C). Convergence to the parameter of interest, otherwise known as consistency, gives a theoretical justification for an estimator. Below we show that $\hat{r}_{\mathrm{ER}}^2$ is consistent for $r_{\mathrm{ER}}^2$ while $\hat{r}^2$ is not.

We note that the covariance in Eq 17 can be removed by using separate subsets of the data for the estimation of $\hat{C}_{\mathrm{ER}_m}$ and $\hat{V}_{\mathrm{ER}_m}$. This leaves the inflation by Jensen's inequality $\mathrm{E}[\frac{1}{\hat{V}_{\mathrm{ER}_m}}] \geq \frac{1}{\mathrm{E}[\hat{V}_{\mathrm{ER}_m}]}$, which could be estimated and corrected for via a simulation-based method such as the parametric bootstrap (see Discussion, "Further work").

**Consistency of $\hat{r}^2_{\mathrm{ER}}$ in $m$.** We aim to show that $\hat{r}^2_{\mathrm{ER}}$ is consistent for $r^2_{\mathrm{ER}}$ in $m$, more formally:

$$\hat{r}^2_{\mathrm{ER}} \xrightarrow{p} r^2_{\mathrm{ER}} \equiv \lim_{m\to\infty} P(|\hat{r}^2_{\mathrm{ER}} - r^2_{\mathrm{ER}}| \geq \epsilon) = 0.$$

We make use of the continuous mapping theorem that guarantees if a random vector $X_m \xrightarrow{p} \vec{c}$, then for a continuous transformation $g$, $g(X_m) \xrightarrow{p} g(\vec{c})$ where the random vector is almost surely different from any discontinuity points. Taking our random vector to be, $[\hat{C}_{\mathrm{ER}_m}, \hat{V}_{\mathrm{ER}_m}]^T$, and our continuous transformation to be, $g([\hat{C}_{\mathrm{ER}_m}, \hat{V}_{\mathrm{ER}_m}]^T) = \frac{\hat{C}_{\mathrm{ER}_m}}{\hat{V}_{\mathrm{ER}_m}} = \hat{r}^2_{\mathrm{ER}}$ (assuming expectation of the denominator is non-zero), it then suffices to show that $\hat{C}_{\mathrm{ER}_m}$ and $\hat{V}_{\mathrm{ER}_m}$ themselves are consistent estimators for the numerator and denominator of $r^2_{\mathrm{ER}}$.

First, we have already shown that $\hat{C}_{\mathrm{ER}_m}$ and $\hat{V}_{\mathrm{ER}_m}$ are unbiased estimators. Next, we must show that their variance is decreasing with $m$, and then via Chebyshev's inequality,

$$P(|X - \mu| \geq \epsilon) \leq \frac{\mathrm{Var}[X]}{\epsilon^2},$$

we can show their convergence to their expectation. Here we consider the case where $\hat{\sigma}^2 = s^2$. Since the model predictions ($v_i$) are fixed for the purpose of the proof, we assume the dot product between model predictions and neural responses is scaled linearly by $m$:

$$\frac{1}{m}\left(\sum_i^m (v_i - \bar{v})(\mu_i - \bar{\mu})\right)^2 = c,$$

as is the dynamic range of the neuron:

$$\frac{1}{m}\sum_{i=1}^m (\mu_i - \bar{\mu})^2 = v,$$

and we scale the numerator and denominator of $\hat{r}^2_{\mathrm{ER}}$ by $\frac{1}{m}$ which makes no change to their ratio.

The numerator, $\hat{C}_{\mathrm{ER}_m} = \frac{1}{m}\left[\left(\sum_i^m (v_i - \bar{v})(Y_i - \bar{Y})\right)^2 - \frac{s^2}{n}\right]$, has variance equal to the sum of the variance of its first and second term (since they are independent). Since $\mathrm{Var}[\chi^2_m(\lambda)] = 2m + 4\lambda$ the variances are, respectively,

$$\mathrm{Var}\left[\frac{1}{m}\left(\sum_{i=1}^m (v_i - \bar{v})(Y_i - \bar{Y})\right)^2\right] = \mathrm{Var}\left[\frac{\sigma^2}{nm}\chi^2_1\left(\frac{\left(\sum_{i=1}^m (v_i - \bar{v})(\mu_i - \bar{\mu})\right)^2}{\sigma^2/n}\right)\right]$$

$$= \frac{2\sigma^4}{n^2 m^2} + \frac{4\sigma^2 c}{mn},$$

and

$$\mathrm{Var}\left[\frac{s^2}{nm}\right] = \frac{1}{n^2 m^2}\frac{2\sigma^4}{mn - 1},$$

thus

$$\mathrm{Var}[\hat{C}_{\mathrm{ER}_m}] = \frac{2\sigma^4}{n^2 m^2} + \frac{4\sigma^2 c}{mn} + \frac{1}{n^2 m^2}\frac{2\sigma^4}{mn - 1}.$$

The denominator, $\hat{V}_{\mathrm{ER}_m} = \frac{1}{m}\left(\sum_{i=1}^{m}\left(Y_i - \bar{Y}\right)^2 - (m-1)\frac{s^2}{n}\right)$, also has variance equal to the sum of the variance of its first and second term (by independence). The variances are, respectively,

$$\mathrm{Var}\left[\frac{1}{m}\sum_{i=1}^{m}\left(Y_i - \bar{Y}\right)^2\right] = \mathrm{Var}\left[\frac{\sigma^2}{nm}\chi_{m-1}^2\left(\frac{\sum_{i=1}^{m}\left(\mu_i - \bar{\mu}\right)^2}{\frac{\sigma^2}{n}}\right)\right] = \frac{2\sigma^4(m-1)}{n^2 m^2} + \frac{4\sigma^2 v}{mn},$$

and

$$\mathrm{Var}\left[\frac{(m-1)}{nm}s^2\right] = \frac{(m-1)}{nm}\frac{\sigma^4}{mn-1},$$

thus

$$\mathrm{Var}[\hat{V}_{\mathrm{ER}_m}] = \frac{2\sigma^4(m-1)}{n^2 m^2} + \frac{4\sigma^2 v}{mn} + \frac{(m-1)}{nm}\frac{\sigma^4}{mn-1}.$$

For both $\mathrm{Var}[\hat{V}_{\mathrm{ER}_m}]$ and $\mathrm{Var}[\hat{C}_{\mathrm{ER}_m}]$, all but $m$ is constant; therefore, we can find an $m$ to scale variance below any given $\epsilon$. So by Chebyshev's inequality we have:

$$P(|\hat{C}_{\mathrm{ER}_m} - c| \geq \epsilon) \leq \frac{\mathrm{Var}[\hat{C}_{\mathrm{ER}_m}]}{\epsilon^2}.$$

Since as $m \to \infty$, $\frac{\sigma^2_{\hat{C}_{\mathrm{ER}_m}}}{m\epsilon^2} \to 0$ we have that,

$$\lim_{m\to\infty} P(|\hat{C}_{\mathrm{ER}_m} - c| > \epsilon) = 0 \equiv \hat{C}_{\mathrm{ER}_m} \xrightarrow{p} c,$$

and similarly,

$$\hat{V}_{\mathrm{ER}_m} \xrightarrow{p} v.$$

Thus by the continuous mapping theorem:

$$\hat{r}_{\mathrm{ER}}^2 = \frac{\hat{C}_{\mathrm{ER}_m}}{\hat{V}_{\mathrm{ER}_m}} \xrightarrow{p} \frac{c}{v} = r_{\mathrm{ER}}^2.$$

In contrast, we show below that the naive estimator is not consistent and provide insight into when the difference between $\hat{r}^2$ and $\hat{r}_{\mathrm{ER}}^2$ is large.

**Inconsistency of $\hat{r}^2$ in $m$.**   Similarly to the previous derivation, we can take the numerator and denominator of $\hat{r}^2$ (Eqs 9 and 12), scale by $\frac{1}{m}$, find their expected values, and in turn find the asymptotic value of $\hat{r}^2$. Here, though, we simplify by setting the model to be unit length,

$$\hat{r}_m^2 \xrightarrow{p} \frac{c}{v + \frac{\sigma^2}{n}} \leq \frac{c}{v}.$$

This result shows that $\hat{r}^2$ is not a consistent estimator in $m$ of $r_{\mathrm{ER}}^2$.

## Confidence intervals

Here we develop and prove a method that provides $\alpha$-level confidence intervals for the estimator $\hat{r}_{\mathrm{ER}}^2$. We considered the typical parametric bootstrap and non-parametric bootstrap approaches, but found that they were not reliable for typical ranges of parameters (see Results, "Confidence intervals for $\hat{r}_{\mathrm{ER}}^2$").

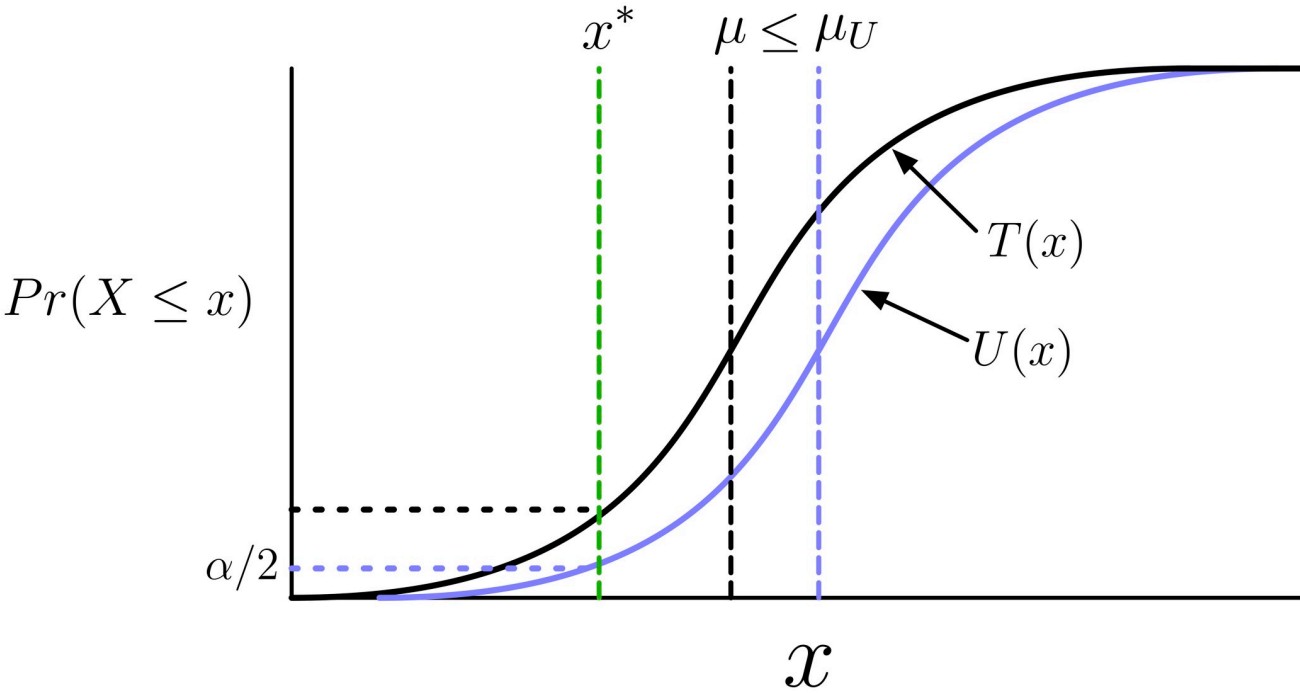

**Fig 14. Illustrative schematic of confidence interval estimation.** Given an observed estimate $x^*$ (green dashed vertical) from the distribution of the estimator $X$ with CDF $T(x)$ (solid black curve) associated with the parameter being estimated $\mu$ (black dashed vertical), the upper limit of the $\alpha$-level confidence interval is the $\mu_U$ (purple vertical dashed) corresponding to the cumulative distribution of $X_U$, $U(x)$ (solid purple curve) that would generate values less than $x^*$ with probability $\alpha/2$ (purple horizontal dashed). Thus $U(x)$ is defined by $U(x^*) = \alpha/2$. Under the assumption the family of CDFs of $X$ are stochastically increasing in $\mu$, the event that $T(x) \geq \alpha/2$ corresponds to the event that $\mu < \mu_U$, thus the upper limit of the confidence interval contains the true value of $\mu$. In graphical terms, if the black horizontal dashed line is above the purple, then it is guaranteed that the purple vertical dashed is to the right of the black. Thus these two events have the same probability: $Pr(\mu \leq \mu_U) = Pr(\alpha/2 \leq T(X)) = 1 - \alpha/2$. Here we have used generic symbols for illustrative purposes, but for reference to the proof (see Methods, "Proof of $\alpha$-level confidence intervals"), the notation used here correspond as follows: $X = \hat{r}_{ER}^2$, $x^* = \hat{r}_{ER}^{2*}$, $\mu = r_{ER}^2$, $\mu_U = r_{ER(h)}^2$, $T(x) = F(\hat{r}_{ER}^2 | r_{ER}^2)$, and $U(x) = F(\hat{r}_{ER}^2 | r_{ER(h)}^2)$.

Our approach hinges upon finding the lowest $r_{ER}^2$ whose distribution would give an estimate greater than the observed $\hat{r}_{ER}^{2*}$ with probability $\alpha/2$, calling this $r_{ER(l)}^2$, and finding the highest $r_{ER}^2$ that would give an estimate less then the observed $\hat{r}_{ER}^{2*}$ with probability $\alpha/2$, calling this $r_{ER(h)}^2$. The interval $[r_{ER(l)}^2, r_{ER(h)}^2]$ then serves as our $\alpha$- level confidence interval (see Fig 14 for graphical explanation). We use a Bayesian framework to sample from the probability distribution, $f(\hat{r}_{ER}^2 | r_{ER}^2)$, parameterized by the observed neural statistics $s^2$ and $\hat{d}^2$, allowing us to find $[r_{ER(l)}^2, r_{ER(h)}^2]$ under assumed uninformative priors on $\sigma^2$ and $d^2$ (see "Computing confidence intervals", below).

**Proof of $\alpha$-level confidence intervals.** Here, we justify this procedure for the case of $r_{ER(h)}^2$ ($r_{ER(l)}^2$ is similar). Our two main assumptions are that the cumulative distribution $F(\hat{r}_{ER}^2 | r_{ER}^2)$ is stochastically increasing in $r_{ER}^2$,

$$r_{ER}^2 \leq r_{ER}^{2'} \Leftrightarrow F(\hat{r}_{ER}^2 | r_{ER}^2) \geq F(\hat{r}_{ER}^2 | r_{ER}^{2'}), \tag{18}$$

and that we can always find an $r_{ER}^2$ such that for any observed $\hat{r}_{ER}^{2*}$,

$$F(\hat{r}_{ER}^{2*} | r_{ER}^2) = \alpha \in (0, 1). \tag{19}$$

We now consider two mutually exclusive possibilities. First, with probability $1 - \alpha/2$, the

observed $\hat{r}^{2*}_{\text{ER}}$ is large enough to satisfy:

$$F(\hat{r}^{2*}_{\text{ER}}|r^2_{\text{ER}}) \geq \alpha/2.$$

Then by the assumption in Eq 19 we can find a $r^2_{\text{ER}(h)}$ where,

$$F(\hat{r}^{2*}_{\text{ER}}|r^2_{\text{ER}(h)}) = \alpha/2,$$

and under our initial assumption (Eq 18), this implies

$$r^2_{\text{ER}} \leq r^2_{\text{ER}(h)},$$

because

$$F(\hat{r}^{2*}_{\text{ER}}|r^2_{\text{ER}}) \geq \alpha/2 = F(\hat{r}^{2*}_{\text{ER}}|r^2_{\text{ER}(h)}).$$

Second, if on the other hand $\hat{r}^{2*}_{\text{ER}}$ is small enough such that

$$F(\hat{r}^{2*}_{\text{ER}}|r^2_{\text{ER}}) < \alpha/2,$$

then

$$r^2_{\text{ER}} > r^2_{\text{ER}(h)}.$$

Thus, under repeated sampling, with the desired probability $\alpha/2$, the upper limit of our confidence interval, $r^2_{\text{ER}(h)}$, does not contain $r^2_{\text{ER}}$. The proof for the lower end of the confidence interval $r^2_{\text{ER}(l)}$ is similar. The probability of the mutually exclusive events that either $r^2_{\text{ER}} > r^2_{\text{ER}(h)}$ or $r^2_{\text{ER}} < r^2_{\text{ER}(l)}$ is the sum of the probability of the two events, $\alpha$. See Fig 14 for a graphical explanation of this proof.

For simplicity of the proof, we assumed that it was possible to find $F(\hat{r}^{2*}_{\text{ER}}|r^2_{\text{ER}(h)}) = \alpha/2$, which is not necessarily the case because $r^2_{\text{ER}(h)} \in [0, 1]$ is bounded but $\hat{r}^{2*}_{\text{ER}}$ is not. If $F(\hat{r}^{2*}_{\text{ER}}|r^2_{\text{ER}} = 1) > \alpha/2$ or $F(\hat{r}^{2*}_{\text{ER}}|r^2_{\text{ER}} = 0) < \alpha/2$, then there is no $r^2_{\text{ER}(h)}$ that will achieve $\alpha/2$. Under the condition where $F(\hat{r}^{2*}_{\text{ER}}|r^2_{\text{ER}} = 1) > \alpha/2$, we simply set $r^2_{\text{ER}(h)} = 1$, and since $F(\hat{r}^{2*}_{\text{ER}}|r^2_{\text{ER}} = 1) > \alpha/2$ implies $F(\hat{r}^{2*}_{\text{ER}}|r^2_{\text{ER}} \in [0, 1]) > \alpha/2$, the confidence interval will contain the true value.

Under the condition $F(\hat{r}^{2*}_{\text{ER}}|r^2_{\text{ER}} = 0) < \alpha/2$, we set $r^2_{\text{ER}(h)} = 0$, but we must set the confidence interval, though normally inclusive, to be non-inclusive. Intuitively, this is because if $r^2_{\text{ER}} = 0$, then the upper end of the confidence interval would always contain the true value, and we would be restricted to $\alpha = 1$. Making the CI non-inclusive avoids this problem. Doing this does not cause a problem when the true $r^2_{\text{ER}} > 0$, because $F(\hat{r}^{2*}_{\text{ER}}|r^2_{\text{ER}} = 0) < \alpha/2$ implies $F(\hat{r}^{2*}_{\text{ER}}|r^2_{\text{ER}} \in [0, 1]) < \alpha/2$, the confidence interval should not contain the true $r^2_{\text{ER}}$ and it does not because $r^2_{\text{ER}} > 0 = r^2_{\text{ER}(h)}$. The case for $r^2_{\text{ER}(l)}$ is similar.

In summary, our confidence interval is defined to be $[r^2_{\text{ER}(l)}, r^2_{\text{ER}(h)}]$ when $r^2_{\text{ER}(l)} < 1$ and $r^2_{\text{ER}(h)} > 0$ but $\emptyset$ (the empty set) if $r^2_{\text{ER}(l)} = 1$ or $r^2_{\text{ER}(h)} = 0$. The lower bound, $r^2_{\text{ER}(l)}$, satisfies $F(\hat{r}^{2*}_{\text{ER}}|r^2_{\text{ER}(l)}) = 1 - \alpha/2$, except if $F(\hat{r}^{2*}_{\text{ER}}|r^2_{\text{ER}} = 1) > 1 - \alpha/2$ or $F(\hat{r}^{2*}_{\text{ER}}|r^2_{\text{ER}} = 0) < 1 - \alpha/2$, then respectively $r^2_{\text{ER}(l)} = 1$ or $r^2_{\text{ER}(l)} = 0$. The upper bound, $r^2_{\text{ER}(h)}$, satisfies $F(\hat{r}^{2*}_{\text{ER}}|r^2_{\text{ER}(h)}) = \alpha/2$, except if $F(\hat{r}^{2*}_{\text{ER}}|r^2_{\text{ER}} = 1) > \alpha/2$ or $F(\hat{r}^{2*}_{\text{ER}}|r^2_{\text{ER}} = 0) < \alpha/2$, then respectively $r^2_{\text{ER}(h)} = 1$ or $r^2_{\text{ER}(h)} = 0$.

To sample from the conditional distribution $f(\hat{r}^2_{\text{ER}}|s^2, \hat{d}^2, r^2_{\text{ER}})$, we assume that $\sigma^2$ and $d^2$ follow an uninformative non-negative uniform prior ($U[0, \infty]$), and given their observed estimates $s^2$ and $\hat{d}^2$, we obtain samples from the posterior distribution of $\sigma^2$ and $d^2$ via the Metropolis-Hastings sampling method (for details see "Bayesian model and simulation"). For

a chosen $r^2_{ER}$ (e.g., a candidate for $r^2_{ER(h)}$, we sample from $f(\hat{r}^2_{ER}|s^2, \hat{d}^2, r^2_{ER})$ by drawing samples of $\sigma^2$ and $d^2$ from the posterior distribution $f(\sigma^2, d^2|s^2, \hat{d}^2)$ while $r^2_{ER}$ is fixed to the desired value. Thus for each sample we then draw observations $Y$ and predictions $\nu_i$ from the model described in Eq 6 and finally calculate $\hat{r}^2_{ER}$ for a sample from $f(\hat{r}^2_{ER}|s^2, \hat{d}^2, r^2_{ER})$.

**Computing confidence intervals.**   We use a simple iterative bracketing algorithm to narrow down the range of candidate values for the ends of our confidence interval. For example, to estimate $r^2_{ER(h)}$ within [0, 1], we first evaluate the highest possible value: 1. We sample $N = 2,500$ draws from $f(\hat{r}^2_{ER}|s^2, \hat{d}^2, r^2_{ER(c)} = 1)$ to find the proportion, $\hat{p}$, of those less than or equal to $\hat{r}^{2*}_{ER}$. We then calculate a z-statistic to test whether this is significantly different from the desired $\alpha/2$:

$$z = \frac{\hat{p} - \alpha/2}{\hat{p}(1 - \hat{p})/N}.$$

At some desired significance level (here $p < 0.01$), we either do not reject the null and accept $r^2_{ER(h)} = r^2_{ER(c)}$, or we reject the null. In the latter case, if $z$ is positive we determine that $r^2_{ER(h)}$ must be higher, whereas if $z$ is negative it must be lower. In the case where $r^2_{ER(c)} = 1$ and $z$ is positive, there are no higher possible values of $r^2_{ER(h)}$ and thus we accept $r^2_{ER(h)} = r^2_{ER(c)}$. Otherwise, on the next step we choose a new candidate by sampling from $r^2_{ER(c)} \sim U[0, 1]$ then evaluating the result and if we reject the null and $z$ is positive our new interval will be $[r^2_{ER(c)}, 1]$ and if $z$ is negative $[0, r^2_{ER(c)}]$. Otherwise, if we do not reject the null we accept $r^2_{ER(h)} = r^2_{ER(c)}$. We continue this bracketing until we do not reject the null or a pre-determined number of splits has passed (here we use 100). Accuracy of this algorithm will increase with number of splits and simulation samples.

**Confidence interval validation.**   We used simulations to evaluate our confidence interval methods under the sampling distribution $f(\hat{r}^2_{ER}|s^2, \hat{d}^2, r^2_{ER})$. Conceptually, this is the distribution of $\hat{r}^2_{ER}$ after data has been collected and sample variance and sample dynamic range calculated, and now we wish to calculate the data's fit to a model with unknown but fixed $r^2_{ER}$. To demonstrate that our method contains the unknown $r^2_{ER}$ at the desired $\alpha$, our procedure is as follows. For a chosen $n$, $m$, $\sigma^2$, $d^2$, and $r^2_{ER}$, sample an $n \times m$ data matrix (Y) and calculate its sample variance $s^2$ and dynamic range $\hat{d}^2$. Then using the Metropolis-Hastings algorithm (see Methods, "Bayesian model and simulation"), draw 5,000 samples from the posterior distribution $f(\sigma^2, d^2|s^2, \hat{d}^2)$. Next, we simulate the distribution of $\hat{r}^2_{ER}$ for each of these data samples by drawing from $f(\hat{r}^2_{ER}|\sigma^2, d^2, r^2_{ER})$. For each of these draws, we construct confidence intervals, and then we calculate the proportion of times that the confidence intervals contain the true $r^2_{ER}$. This proportion estimates the true $\alpha$ level of the confidence interval method.

**Bayesian model and simulation.**   We sample from the posterior of two parameters: $\sigma^2$ and $d^2 = \frac{1}{m}\sum_{i=1}^{m}(\mu_i - \bar{\mu})^2$. Their associated sufficient statistics are:

$$\hat{s}^2 = \frac{1}{m(n-1)}\sum_{i=1}^{m}\sum_{j=1}^{n}(Y_{i,j} - \bar{Y}_{i,\cdot})^2$$

$$\hat{d}^2 = \frac{1}{m-1}\sum_{i=1}^{m}(\bar{Y}_{i,\cdot} - \bar{Y}_{\cdot,\cdot})^2$$

and their distributions are:

$$\hat{s}^2 \sim \frac{\sigma^2}{m(n-1)} \chi^2_{m(n-1)} \tag{20}$$

$$\hat{d}^2 \sim \frac{\sigma^2}{n(m-1)} \chi^2_{m-1} \left( \frac{\sum_{i=1}^{m} (\mu_i - \bar{\mu})^2}{\frac{\sigma^2}{n}} \right) \tag{21}$$

By Bayes theorem we have,

$$P(\sigma^2, d^2 | \hat{s}^2, \hat{d}^2) \propto P(\hat{s}^2, \hat{d}^2 | \sigma^2, d^2) P(\sigma^2, d^2) = P(\hat{s}^2 | \sigma^2) P(\hat{d}^2 | d^2) \mathbf{1}(\sigma^2)_{[0,\infty]} \mathbf{1}(d^2)_{[0,\infty]},$$

where the equality is derived by recognizing the sample variance ($\hat{s}^2$) and dynamic range ($\hat{d}^2$) are independent and setting the prior to be uniform non-negative. The estimates $\hat{s}^2$ and $\hat{d}^2$ are fixed, calculated from the data, and our goal is to look up the distribution of the parameters given these fixed values. We use the Metropolis-Hastings algorithm to draw from the desired distribution $P(\sigma^2, d^2 | \hat{s}^2, \hat{d}^2)$ and approximate it with the empirical distribution (a histogram). Our sampling procedure is as follows. We initialize our parameter samples $\sigma^2, d^2$ at their estimates $\hat{s}^2, \hat{d}^2$, and we then sample a new candidate from our proposal distribution: a truncated multivariate normal with means $\hat{s}^2, \hat{d}^2$ and diagonal variances equal to the variance of the distributions (Eqs 20 and 21) where $\sigma^2 = \hat{s}^2, d^2 = \hat{d}^2$. We take the ratio of likelihoods,

$$a = P(\hat{s}^2, \hat{d}^2 | \sigma^2_{\text{proposal}}, d^2_{\text{proposal}}) / P(\hat{s}^2, \hat{d}^2 | \sigma^2_{\text{current}}, d^2_{\text{current}}).$$

If $a > 1$, we accept the candidates as our new current samples, but if $a < 1$, we then draw from $u \sim U[0,1]$. If $u < a$, we also accept the candidates but if not, we retain the current samples. Throughout the paper we run the chain for 5,000 iterations then randomly sample with replacement from it.

### SNR relation to F-test and number of trials

Our goal is to be able find for a given SNR and number of repeats the number of stimuli needed to reliably detect tuning under an F-test. To calculate the F-statistic for testing whether there is variation in the expected responses across stimuli (i.e., stimulus selectivity), we form the ratio,

$$F = \frac{\frac{n}{m-1} \sum_{i=1}^{m} \left( \bar{Y}_{i,\cdot} - \bar{Y}_{\cdot,\cdot} \right)^2}{\frac{1}{n(m-1)} \sum_{i=1}^{m} \sum_{j}^{n} \left( Y_{i,j} - \bar{Y}_{i,\cdot} \right)^2},$$

where for clarity we indicate dimensions averaged over with a dot. The numerator calculates the amount of variance explained by stimuli and the denominator calculates the amount of variance unexplained by stimuli. The numerator is a scaled non-central $\chi^2$ distribution:

$$\frac{n}{m-1} \sum_{i=1}^{m} \left( \bar{Y}_{i,\cdot} - \bar{Y}_{\cdot,\cdot} \right)^2 \sim \frac{n}{m-1} \frac{\sigma^2}{n} \chi^2_{m-1} \left( \frac{\sum (\mu_i - \bar{\mu})^2}{\sigma^2/n} \right) = \frac{n}{m-1} \frac{\sigma^2}{n} \chi^2_{m-1}(mn\text{SNR}),$$

where the final equality comes from the definition of SNR (5). The denominator is a central $\chi^2$ distribution:

$$\frac{1}{n(m-1)} \sum_{i=1}^{m} \sum_{j}^{n} \left( Y_{i,j} - \bar{Y}_{i,\cdot} \right)^2 \sim \frac{\sigma^2}{n(m-1)} \chi^2_{m(n-1)}.$$

Thus taking the ratio we have a singly non-central F-distribution:

$$F_{m-1,m(n-1)}(mn\text{SNR}). \tag{22}$$

To test for significant tuning, we set an $\alpha$-level criterion, $c_{F(\alpha)}$, under the null hypothesis that the observed F-statistic is from a central F-distribution:

$$P[F_{m-1,m(n-1)} \geq c_{F(\alpha)}] = \alpha.$$

Finally, given $m$ and $n$, we can find the SNR where, for some high probability $\beta$,

$$P[F_{m-1,m(n-1)}(mn(\text{SNR})) > c_{F(\alpha)}] = \beta.$$

We set $\beta = 0.99$ and $\alpha = 0.01$ and numerically solve for the SNR.

## Electrophysiological data

We reanalyzed a variety of neuronal data from previous studies. This includes three experiments in area V4 and one in MT of the awake, fixating rhesus monkey (Macaca mulattta), as well as spiking and two-photon imaging in awake mouse VISp. Experimental protocols for all studies are described in detail in the original publications.

From Pasupathy and Connor [2], we examined responses of 109 V4 neurons to a set of 362 shapes. There were typically 3-5 repeats of each stimulus, but we used only the 96 cells that had at least 4 repeats for all stimuli. We used the spike count for each trial during the 500 ms stimulus presentation.

From Popovkina et al. [23], we examined responses of 43 V4 neurons (7 from one monkey, 36 from another) to filled stimuli (drawn from the same set of shapes used for the previous study) and to outline stimuli that were the same except the fill was set to be equivalent to background color. Stimulus color and luminance were customized to elicit a robust response from the recorded neuron. Spikes were counted over the 300 ms duration of each stimulus presentation.

From the 2019 UW V4 Neural Data Challenge, we examined single unit (SUA) and multi-unit (MUA) data from 7 V4 recordings. Up to 601 images were shown with between 3-20 repeats for each image. The images were drawn semi-randomly from the 2012 ILSVRC validation set of images [24] where an 80X80 pixel patch was sampled and had a soft window applied (circular Gaussian, SD 16 pixels, applied to the alpha channel). Images were shown for 300 ms with 250 ms in between images. The model we analyze was the winner of the Neural Data Challenge (out of 32 competitors) on held-out data from the 14 sets of V4 responses to natural images.

From Zohary et al. [12, 13], we examined responses from 81 pairs of MT neurons recorded from three awake rhesus monkey (Macaca mulatta) viewing dynamic random dots (stimuli described in Britten et al. [25]). Optimal speed of drifting dots was found for the one of the two neurons being recorded. Eight different directions of motion at 45° increments were repeated 10-20 times. Monkeys performed a two alternative forced choice task of motion direction discrimination during the experiment. Post-stimulus spikes were counted in the 2 s window of stimulus presentation. Experimenters were rigorous in only recording from pairs of neurons whose spike waveforms were strikingly different.

From the Allen Institute for Brain Science (AIBS) mouse database [16], we examined calcium fluorescence data. Fluorescence of mouse visual cortex neurons expressing GCaMP6f was measured via 2-photon imaging through a cranial window. We analyzed signals recorded in response to natural scenes and static gratings presented for 0.25 s each with no interval between with 50 repeats in random order. The natural scene stimulus consisted of

118 natural images from a variety of databases. The static grating stimulus consisted of a full field static sinusoidal grating at a single contrast (80%). Gratings were presented at 6 orientations, 5 spatial frequencies (0.02, 0.04, 0.08, 0.16, 0.32 cycles/degree), and 4 phases (0, 0.25, 0.5, 0.75). For every trial, $\Delta F/F$ was estimated using the average fluorescence of the preceding second (4 image presentations) as baseline. We analyzed the average change in fluorescence during the 1/2 s period after the start of the image presentation relative to 1 s before. We also examined SUA data from the AIBS mouse neuropixel data set [17], which was recorded in response to the same stimuli as the calcium data. Spike counting windows were 0.25 s.

## Prior analytic methods for estimating $r_{ER}^2$

Our estimator, $\hat{r}_{ER}^2$, is derived via the strategy of Haefner and Cumming [6] to unbias the numerator and denominator of the coefficient of determination. Haefner and Cumming in turn cite Sahani and Linden [3] as the predecessor to their method. Sahani and Linden constructed an unbiased estimator of the variation in the expected response of the neuron (i.e. tuning curve) they called this 'signal power'. They normalize an estimator of explained variation, unbiased with respect to noise under conditions they did not specify (1-parameter regression of model predictions, see Methods, "Derivation of Normalized Signal Power Explained (SPE_norm)"). Sahani and Linden, by not specifying how a given model prediction should be fit to the neural data before estimating the quality of the fit, introduced potential problems in their estimator. This was recognized by Schoppe et al. [8], who point out that the estimator was sensitive to differences between the mean and amplitude of the model predictions and the neural data. Consequently, the estimator could give large negative values because the squared error between model predictions and neural responses was unbounded. This criticism, while technically correct, is easily overcome by regressing (with intercept term) the given model predictions onto the neural data before using normalized SPE. Schoppe et al., motivated by the problems they found in SPE, focused on simplifying CC_norm of Hsu et al. [4] to not require re-sampling by making use of the signal power estimator developed by Sahani and Linden. They derived a simple estimator whose square, termed here $CC_{norm-SP}^2$, we find is essentially numerically equivalent to SPE_norm of Sahani and Linden in the case of one-parameter regression.

For the purpose of comparison, below we write out the exact formulas and approximate expected values of two prior methods [3, 6] that are closely related to $\hat{r}_{ER}^2$ in the notation we use throughout our paper. For all estimators, we assume responses to $m$ stimuli with $n$ repeats where variance have been stabilized. The response to the $i$th stimulus, $j$th repeat, is $Y_{i,j} \sim N(\mu_i, \sigma^2)$ where $\sigma^2$ is the trial-to-trial variability and $\mu_i$ the $i$th expected value of response after variance stabilization. The predictions are fixed for the $m$ stimuli and the $i$th predicted expected value of the data is $v_i$ and we assume they have been fit by a linear model with $d$ degrees of freedom. When averaging data across trials our notation will be $\bar{Y}_{i,\cdot} = \frac{1}{n}\sum_{j=1}^{n} Y_{i,j}$ and across stimuli $\bar{Y}_{\cdot,j} = \frac{1}{m}\sum_{i=1}^{n} Y_{i,j}$.

**Derivation of normalized signal power explained (SPE_norm).** The original description of SPE [3] did not specify, when calculating sample estimates of 'power' (better known as variance), whether to normalize by $m - 1$ (the unbiased estimate) or $m$ (the MLE). Nor was it specified whether an average across trials was used when calculating the difference between variance of the measured response and the residual. We thus use the formula as described by Schoppe et al. [8], who provide code to unambiguously calculate SPE, albeit in a manner

that differs from their text (in particular, their TP $= (n-1) \sum_{j=1}^{n} \frac{1}{m-1} \sum_{i=1}^{m} \left( Y_{i,j} - \bar{Y}_{\cdot,j} \right)^2$)
(similar to Eq 5 of Sahani and Linden), but in their code it is implemented as
$\frac{1}{n} \sum_{j=1}^{n} \frac{1}{m-1} \sum_{i=1}^{m} \left( Y_{i,j} - \bar{Y}_{\cdot,j} \right)^2$)). In the context of the quantities being estimated, the latter
makes more sense (see calculation of expected value of denominator below). For comparison
to the derivation in Schoppe et al., their notation equates to ours as follows: $N = n$, $T = m$,
$R = Y_{i,j}$, $y = \bar{Y}_{i,\cdot}$, $\hat{y} = \hat{v}_i$, and $Var(y) = \frac{1}{m-1} \sum_{i=1}^{m} \left( \bar{Y}_{i,\cdot} - \bar{Y}_{\cdot,\cdot} \right)^2$. Finally, in our notation their esti-
mator is:

$$
\text{SPE}_{\text{norm}} = \frac{\frac{1}{m-1} \sum_{i=1}^{m} \left( \bar{Y}_{i,\cdot} - \bar{Y}_{\cdot,\cdot} \right)^2 - \frac{1}{m-1} \sum_{i=1}^{m} \left( \bar{Y}_{i,\cdot} - \hat{v}_i \right)^2}{\frac{1}{n-1} \left( n \frac{1}{m-1} \sum_{i=1}^{m} \left( \bar{Y}_{i,\cdot} - \bar{Y}_{\cdot,\cdot} \right)^2 - \frac{1}{n} \sum_{j=1}^{n} \frac{1}{m-1} \sum_{i=1}^{m} \left( Y_{i,j} - \bar{Y}_{\cdot,j} \right)^2 \right)}.
$$

Calculating the expectation of the numerator and the denominator for the fit of a linear
model with $d$ degrees of freedom, we can find the asymptotic expectation. Numerator:

$$
\begin{aligned}
&\mathrm{E}\left[ \frac{1}{m-1} \sum_{i=1}^{m} (\bar{Y}_{i,\cdot} - \bar{Y}_{\cdot,\cdot})^2 - \frac{1}{m-1} \sum_{i=1}^{m} (\bar{Y}_{i,\cdot} - \hat{v}_i)^2 \right] \\
&= \frac{1}{m-1} \left( \mathrm{E}\left[ \sum_{i=1}^{m} (\bar{Y}_{i,\cdot} - \bar{Y}_{\cdot,\cdot})^2 \right] - \mathrm{E}\left[ \sum_{i=1}^{m} (\bar{Y}_{i,\cdot} - \hat{v}_i)^2 \right] \right) \\
&= \frac{1}{m-1} \left( \mathrm{E}\left[ \frac{\sigma^2}{n} \chi_{m-1}^2 \left( \frac{\sum_{i=1}^{m} (\mu_i - \bar{\mu}_\cdot)^2}{\frac{\sigma^2}{n}} \right) \right] - \mathrm{E}\left[ \frac{\sigma^2}{n} \chi_{m-d}^2 \left( \frac{\sum_{i=1}^{m} (\mu_i - \hat{v}_i)^2}{\frac{\sigma^2}{n}} \right) \right] \right) \\
&= \frac{1}{m-1} \left( \left( \sum_{i=1}^{m} (\mu_i - \bar{\mu}_\cdot)^2 + (m-1) \frac{\sigma^2}{n} \right) - \left( \sum_{i=1}^{m} (\mu_i - \hat{v}_i)^2 + (m-d) \frac{\sigma^2}{n} \right) \right) \\
&= \frac{1}{m-1} \left( \sum_{i=1}^{m} (\mu_i - \bar{\mu}_\cdot)^2 - \sum_{i=1}^{m} (\mu_i - \hat{v}_i)^2 + \frac{\sigma^2}{n} (d-1) \right).
\end{aligned}
$$

Denominator:

$$
\begin{aligned}
&\mathrm{E}\left[ \frac{1}{n-1} \left( n \frac{1}{m-1} \sum_{i=1}^{m} (\bar{Y}_{i,\cdot} - \bar{Y}_{\cdot,\cdot})^2 - \frac{1}{n} \sum_{j=1}^{n} \frac{1}{m-1} \sum_{i=1}^{m} (Y_{i,j} - \bar{Y}_{\cdot,j})^2 \right) \right] \\
&= \frac{1}{n-1} \left( n \frac{1}{m-1} \left( \sum_{i=1}^{m} (\mu_i - \bar{\mu}_\cdot)^2 + (m-1) \frac{\sigma^2}{n} \right) \right. \\
&\qquad \left. - \frac{1}{n} \sum_{j=1}^{n} \frac{1}{m-1} \mathrm{E}\left[ \sigma^2 \chi_{m-1}^2 \left( \frac{\sum_{i=1}^{m} (\mu_i - \bar{\mu}_\cdot)^2}{\sigma^2} \right) \right] \right) \\
&= \frac{1}{n-1} \left( n \frac{1}{m-1} \left( \sum_{i=1}^{m} (\mu_i - \bar{\mu}_\cdot)^2 + (m-1) \frac{\sigma^2}{n} \right) \right. \\
&\qquad \left. - \frac{1}{m-1} \mathrm{E}\left[ \sigma^2 \chi_{m-1}^2 \left( \frac{\sum_{i=1}^{m} (\mu_i - \bar{\mu}_\cdot)^2}{\sigma^2} \right) \right] \right) \\
&= \frac{1}{n-1} \left( n \frac{1}{m-1} \left( \sum_{i=1}^{m} (\mu_i - \bar{\mu}_\cdot)^2 + (m-1) \frac{\sigma^2}{n} \right) \right. \\
&\qquad \left. - \frac{1}{m-1} \left( \sum_{i=1}^{m} (\mu_i - \bar{\mu}_\cdot)^2 + (m-1)\sigma^2 \right) \right) \\
&= \frac{1}{m-1} \frac{1}{n-1} (n-1) \sum_{i=1}^{m} (\mu_i - \bar{\mu}_\cdot)^2 = \frac{1}{m-1} \sum_{i=1}^{m} (\mu_i - \bar{\mu}_\cdot)^2.
\end{aligned}
$$

Putting the expectations into the numerator and denominator we have:

$$\frac{\sum_{i=1}^{m} (\mu_i - \bar{\mu}_.)^2 - \sum_{i=1}^{m} (\mu_i - \hat{v}_i)^2 + \frac{\sigma^2}{n}(d-1)}{\sum_{i=1}^{m} (\mu_i - \bar{\mu}_.)^2}.$$

We note that only if $d = 1$ (i.e., the model has only 1 term) is the numerator unbiased. Below we describe how Haefner and Cumming developed an estimator that accounts for degrees of freedom more generally.

**Derivation of $\Upsilon$.** For comparison to the original paper of Haefner and Cumming [6], we give their notation and its equivalent terms in our notation: $d_{i,j} = Y_{i,j}$, $d_i = \bar{Y}_{i,\cdot}$, $\bar{\bar{d}} = \bar{Y}_{\cdot,\cdot}$, $\Sigma^2 = \sigma^2$, $N = m$, $N_\sigma = m(n-1)$, $R = n$, $n = d$, $D_i = \mu_i$, $M_i = v_i$, $m_i = \hat{v}_i$, $\sigma^2 = \hat{\sigma}^2$, $\lambda_{DD} = \sum_{i=1}^{m} (\mu_i - \bar{\mu}_.)^2/\sigma^2$, $\lambda_{DM} = \sum_{i=1}^{m} (\mu_i - v_i)^2$.

These authors explicitly attempt to remove the bias of the coefficient of determination:

$$r^2 \equiv 1 - \frac{\sum_{i=1}^{m} (\bar{Y}_{i,\cdot} - v_i)^2}{\sum_{i=1}^{m} (\bar{Y}_{i,\cdot} - \bar{Y}_{\cdot,\cdot})^2}.$$

Their unbiased estimator is derived by dividing the numerator and denominator by the sample trial-to-trial variability ($\hat{\sigma}^2 = \frac{1}{m}\sum_{i=1}^{m} \frac{1}{n-1}\sum_{j=1}^{n} (Y_{i,j} - \bar{Y}_{\cdot,j})^2$) and their respective degrees of freedom ($d$ below being the degrees of freedom of the linear model), noting the numerator and denominator become non-central F-distributions, then shifting and scaling these to provide unbiased estimates. Since $\mathrm{E}[F_{d_1,d_2}(\lambda)] = \frac{d_2(d_1+\lambda)}{d_1(d_2-2)}$, the expectation of the numerator is,

$$\mathrm{E}\left[\frac{1}{m-d}\sum_{i=1}^{m}(\bar{Y}_{i,\cdot} - \hat{v}_i)^2/\hat{\sigma}^2\right] = \mathrm{E}\left[F_{(m-d),m(n-1)}\left(\sum_{i=1}^{m}\frac{(\mu_i - \hat{v}_i)^2}{\sigma^2}\right)\right]$$

$$= \frac{m(n-1)(m-d+\sum_{i=1}^{m}(\mu_i - \hat{v}_i)^2/\sigma^2)}{(m-d)(m(n-1)-2)},$$

thus the unbiased estimate of the numerator is:

$$\mathrm{E}\left[\frac{(m-d)(m(n-1)-2)}{m(n-1)}\left(\frac{1}{m-d}\sum_{i=1}^{m}(\bar{Y}_{i,\cdot} - \hat{v}_i)^2\right)/\hat{\sigma}^2 - (m-d)\right] = \sum_{i=1}^{m}(\mu_i - \hat{v}_i)^2/\sigma^2.$$

The expectation of the denominator is:

$$\mathrm{E}\left[\frac{1}{m-1}\sum_{i=1}^{m}(\bar{Y}_{i,\cdot} - \bar{Y}_{\cdot,\cdot})^2/\hat{\sigma}^2\right] = \mathrm{E}\left[F_{(m-1),m(n-1)}\left(\sum_{i=1}^{m}\frac{(\mu_i - \bar{\mu}_.)^2}{\sigma^2}\right)\right]$$

$$= \frac{m(n-1)(m-1+\sum_{i=1}^{m}(\mu_i - \bar{\mu}_.)^2/\sigma^2)}{(m-1)(m(n-1)-2)},$$

thus the unbiased estimate of the denominator is:

$$\mathrm{E}\left[\frac{(m-1)(m(n-1)-2)}{m(n-1)}\left(\frac{1}{m-1}\sum_{i=1}^{m}(\bar{Y}_{i,\cdot} - \bar{Y}_{\cdot,\cdot})^2\right)/\hat{\sigma}^2 - (m-1)\right] = \sum_{i=1}^{m}(\mu_i - \bar{\mu}_.)^2/\sigma^2.$$

Forming the ratio, we obtain the Haefner and Cumming estimator:

$$
\begin{aligned}
\Upsilon \quad &= 1 - \frac{\frac{(m-d)(m(n-1)-2)}{m(n-1)}\left(\frac{1}{m-d}\sum_{i=1}^{m}\left(\bar{Y}_{i,\cdot} - \hat{v}_i\right)^2/\hat{\sigma}^2\right) - (m-d)}{\frac{(m-1)(m(n-1)-2)}{m(n-1)}\left(\frac{1}{m-1}\sum_{i=1}^{m}\left(\bar{Y}_{i,\cdot} - \bar{Y}_{\cdot,\cdot}\right)^2/\hat{\sigma}^2\right) - (m-1)} \\
&= 1 - \frac{\sum_{i=1}^{m}\left(\bar{Y}_{i,\cdot} - \hat{v}_i\right)^2/\hat{\sigma}^2 - \frac{m(n-1)}{m(n-1)-2}(m-d)}{\sum_{i=1}^{m}\left(\bar{Y}_{i,\cdot} - \bar{Y}_{\cdot,\cdot}\right)^2/\hat{\sigma}^2 - \frac{m(n-1)}{m(n-1)-2}(m-1)}.
\end{aligned}
$$

## Extension of $\hat{r}_{\mathrm{ER}}^2$ to fit of linear model

The derivation of Haefner and Cumming via the non-central $F$ was not necessary: the expectation of the numerator and denominator are straightforward to calculate as non-central $\chi^2$ random variables. While our $\hat{r}_{\mathrm{ER}}^2$ is explicitly meant to be the analogue of Pearson's $r^2$, we re-derive the Haefner and Cumming formula along the lines of $\hat{r}_{\mathrm{ER}}^2$ for measuring the fit of a linear model. We specifically avoid the non-central F-distribution so that it is unnecessary to estimate variance (if for example there is a strong prior for the variance and/or multiple trials were not collected). We assume, as did Haefner and Cumming that $\hat{v}_i$ were fit from a linear model via least squares with $d$ coefficients. The expectation of the numerator is:

$$
\mathrm{E}\left[\sum_{i=1}^{m}\left(\bar{Y}_{i,\cdot} - \hat{v}_i\right)^2\right] = \mathrm{E}\left[\frac{\sigma^2}{n}\chi_{m-d}^2\left(\sum_{i=1}^{m}\frac{(\mu_i - \hat{v}_i)^2}{\frac{\sigma^2}{n}}\right)\right] = \sum_{i=1}^{m}(\mu_i - \hat{v}_i)^2 + (m-d)\frac{\sigma^2}{n},
$$

thus its unbiased estimate is:

$$
\mathrm{E}\left[\sum_{i=1}^{m}\left(\bar{Y}_{i,\cdot} - \hat{v}_i\right)^2 - (m-d)\frac{s^2}{n}\right] = \sum_{i=1}^{m}(\mu_i - \hat{v}_i)^2.
$$

The expectation of the denominator is:

$$
\mathrm{E}\left[\sum_{i=1}^{m}\left(\bar{Y}_{i,\cdot} - \bar{Y}_{\cdot,\cdot}\right)^2\right] = \mathrm{E}\left[\frac{\sigma^2}{n}\chi_{m-1}^2\left(\sum_{i=1}^{m}\frac{(\mu_i - \bar{\mu})^2}{\frac{\sigma^2}{n}}\right)\right] = \sum_{i=1}^{m}(\mu_i - \bar{\mu})^2 + (m-1)\frac{\sigma^2}{n},
$$

thus its unbiased estimate is:

$$
\mathrm{E}\left[\sum_{i=1}^{m}\left(\bar{Y}_{i,\cdot} - \bar{Y}_{\cdot,\cdot}\right)^2 - (m-1)\frac{s^2}{n}\right] = \sum_{i=1}^{m}(\mu_i - \bar{\mu}_{\cdot})^2.
$$

Thus their ratio forms:

$$
\hat{r}_{\mathrm{ER}}^2 = 1 - \frac{\sum_{i=1}^{m}\left(\bar{Y}_{i,\cdot} - \hat{v}_i\right)^2 - (m-d)\frac{\hat{\sigma}^2}{n}}{\sum_{i=1}^{m}\left(\bar{Y}_{i,\cdot} - \bar{Y}_{\cdot,\cdot}\right)^2 - (m-1)\frac{\hat{\sigma}^2}{n}}. \tag{23}
$$

## Acknowledgments

We thank Polina Zamarashkina for collecting data for the UWNDC dataset. We thank Anitha Pasupathy, Dina Popovkina, and the Allen Institute for sharing data. We thank Greg Horwitz and Yen-Chi Chen for helpful suggestions and advice. We thank the participants in the UWNDC, including the winner Oleg Polosin, for their neural models.

## Author Contributions

**Conceptualization:** Dean A. Pospisil.

**Data curation:** Dean A. Pospisil, Wyeth Bair.

**Formal analysis:** Dean A. Pospisil.

**Funding acquisition:** Dean A. Pospisil, Wyeth Bair.

**Investigation:** Dean A. Pospisil.

**Methodology:** Dean A. Pospisil, Wyeth Bair.

**Project administration:** Wyeth Bair.

**Resources:** Wyeth Bair.

**Supervision:** Wyeth Bair.

**Visualization:** Dean A. Pospisil.

**Writing – original draft:** Dean A. Pospisil.

**Writing – review & editing:** Dean A. Pospisil, Wyeth Bair.

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
