## [Decision Letter · Decision Letter 0]

17 May 2021

Dear Mr Pospisil,

Thank you very much for submitting your manuscript "The unbiased estimation of the fraction of variance explained by a model" for consideration at PLOS Computational Biology. As with all papers reviewed by the journal, your manuscript was reviewed by members of the editorial board and by several independent reviewers. The reviewers appreciated the attention to an important topic. Based on the reviews, we are likely to accept this manuscript for publication, providing that you modify the manuscript according to the review recommendations.

Dear Dean and Wyeth,

First, I am very sorry this took so long! Not only was it difficult to find reviewers but two of the reviewers that originally accepted to review never returned their review. We are going to proceed with the two reviews that you see here. Both reviews bring up "major"/"Main" issues but I believe that you will be able to address these. The comparisons with the other metrics with real data as suggested by reviewer2 might not only be illustrative but be a good selling point - no?. I actually believe that some of these other metrics are equivalent to yours mathematically. But you provide a nice statistical foundation and confidence intervals that I don't believe was presented in prior work (although I did not look at all these papers).

Also it would be VERY useful to have this code in a notebook published on git. Maybe this is already the case - but as reviewer 2 I did not see that in the manuscript.

Best wishes,

Frederic.

Sincerely,

Frédéric E. Theunissen

Associate Editor

PLOS Computational Biology

Lyle Graham

Deputy Editor

PLOS Computational Biology

[LINK]

Dear Dean and Wyeth,

First, I am very sorry this took so long! Not only was it difficult to find reviewers but two of the reviewers that originally accepted to review never returned their review. We are going to proceed with the two reviews that you see here. Both reviews bring up "major"/"Main" issues but I believe that you will be able to address these. The comparisons with the other metrics with real data as suggested by reviewer2 might not only be illustrative but be a good selling point - no?. I actually believe that some of these other metrics are equivalent to yours mathematically. But you provide a nice statistical foundation and confidence intervals that I don't believe was presented in prior work (although I did not look at all these papers).

Also it would be VERY useful to have this code in a notebook published on git. Maybe this is already the case - but as reviewer 2 I did not see that in the manuscript.

Best wishes,

Frederic.

Reviewer's Responses to Questions

**Comments to the Authors:**

Reviewer #1: This work focuses on obtaining a consistant, unbiased estimator for the "expected response" correlation. The authors argue, correctly, that the model fit should be judged independent of noise effects. Via an approximation they determine that the numerator and denominator of the correlation calculation both have noise-related biases that the authors propose to estimate and subtract. The authors assess the various properties of this estimator and compare on a number of datasets.

The method itself seems sound and well-presented. I have a number of comments that I believe the authors should consider before publication:

Main points:

1) At the very base, this correlation correction is based on having a single trial prediction of neural activity. For the mean of this distribution to be meaningful, the model prediction must, in some sense be unimodal (no probabilistic models that might offer two scenarios, i.e., P(neuron not active this trial) vs P(activity | neuron is active this trial)). Such effects can be caused by missed place fields, for example, and have are better explained in terms of population activity. Can a version of this metric still apply to such scenarios? This would be important as the field continues to move in the direction of population activity.

2) Modeling of neural spiking statistics is an evolving area. The authors introduce a variance-normalized spiking which can handle specific forms of over- and under- dispersion, however I'm curious if the proposed metric can apply to more general statistical models developed over the past decade. For example, Goris et al. 2014 [R1] show a quadratic increase in overdispersion with rate, and follow-up work has further explored more complex relationships in the mean-variance space [R2]. How would the double stochasticity in such models be handeled? The noise is not a clear-cut addition to the numerator and denominator, and so cannot be simply subtracted due to the cross-terms.

3) I find the discussion on SNR on page 12 a little confusing (Par. starting on line 364). It's not always possible to increase the bin size as there are limiting factors on the chang in the underlying homogeneous spike rate. At some point the rate then becomes meaningless, i.e., you get a sinple "tuning curve" with no time information on the response. This is especially going against the increased emphasis on neural dynamics in the population level that is becoming prominant.

4) The calcium fluorescence experiments requires more detail as to how the model "mean" changes. It has been observed both in real data [R3] and in simulation [R4] that single spikes sometimes do not cause significant changes in the DF/F signal. This is a skewed version of the spike count, and so is the idea here that there is a different statisitcal model for the data that no longer relies on the count of firing events? For example, a linear model on the DF/F changes is possible, but so is a traditional tuning curve model with a conditional probabilistic model over DF/F [R5]. I think the authors mean the former, but this should be clarified mathematically as to what statistical model of "neural firing" is used.

5) The demonstration of inconsistancy in r^2 and consistancy in r_ER^2 don't give much of a window into the nonasyptotic biases. For example, while demonstrating how r^2 is bias even in the limit is interesting, it would be interesting to understand as a function of m how fast r_ER^2 converges. Figure 3 seems to indicate a very slow (linear or sublinear?) convergence rate. Is there any way around this given that typical experiments might not have m>1000 stimuli?

[R1] Goris, Robbe LT, J. Anthony Movshon, and Eero P. Simoncelli. "Partitioning neuronal variability." Nature neuroscience 17.6 (2014): 858-865.

[R2] Charles, Adam S., et al. "Dethroning the Fano Factor: a flexible, model-based approach to partitioning neural variability." Neural computation 30.4 (2018): 1012-1045.

[R3] Wei, Ziqiang, et al. "A comparison of neuronal population dynamics measured with calcium imaging and electrophysiology." PLoS computational biology 16.9 (2020): e1008198.

[R4] Charles, Adam S., et al. "Neural anatomy and optical microscopy (NAOMi) simulation for evaluating calcium imaging methods." bioRxiv (2019): 726174.

[R5] Ganmor, Elad, et al. "Direct estimation of firing rates from calcium imaging data." arXiv preprint arXiv:1601.00364 (2016).

Reviewer #2: This manuscript describes a new method for measuring the performance of models for neural data, accounting for noise in the test set used to evaluate the model without bias. Using simulation, the authors argue that their new estimator is as accurate as or more accurate than previously published estimators designed for the same purpose. They also show that they can measure confidence intervals reliably, which enables identification of neurons with model performance above or below chance. They demonstrate the use of the estimator on several datasets and use it to show that neuronal signal-to-noise provides a critical limitation on model performance.

This study tackles a challenging but important problem. As the authors demonstrate, many methods have been proposed to solve this problem, but none is the obvious best choice. Several details of their estimator indicate that it could be adopted as a standard. Overall, the study appears to have been executed carefully and thoughtfully, and it should be of interest to PLoS CB readership. The comparison with multiple previous methods is particularly commendable. At the same time, to be really convincing, the study should address some additional important points to provide compelling evidence that the proposed metric does in fact work as well or better than existing metrics.

Major concerns

1. (p.4) The assumption that variance is constant, or that the data can be readily scaled for variance to become constant, seems reasonable, but it is important to back up this claim with simulations. One becomes particularly concerned in the case of datasets with very sparse spiking activity and many zero responses, which is typical of neural data. This concern could be addressed with a simulation along the lines of Figs. 2-3 but using more realistic spike-like data mimicking responses to a natural or other complex stimulus.

2. The comparison between methods (Fig. 4) is compelling, but it seems important to perform a similar comparison on a real dataset. Do some of the problematic metrics (CCnorm-pb, r2norm-split-sb) show consistently biased results for the real data? Of course, there is no ground truth here, but it should be possible to show if their estimates are consistently different from r_ER.

3. Can code for measuring r_ER and associated confidence intervals be made available? If it was mentioned somewhere in the manuscript, apologies for missing it. While not absolutely necessary, a simple package with a brief tutorial demonstrating use of this metric would be very helpful to readers and go a long way toward getting researchers to adopt the method.

Lesser concerns

L67 / L473. The concept of neuronal SNR as a metric for screening responsive vs. non-responsive neurons is not new. It may be that the current study has developed more rigorous approach to assessing responsiveness with an SNR metric. But many, many studies exclude a subset of neurons based on some SNR metric for the reasons highlighted in this ms.

L176. “cc2_norm” should this be “cc2_norm-split” to match the label in Fig. 4?

L247. Unclear what is different here, exactly.

L269. “typical” Is this the appropriate term here? This unit seems unusually well-described by the sinusoidal model.

L468. “variance explained for a linear model” versus “Pearson’s r2”. It’s not clear what the difference is between these two scenarios. Perhaps it could be spelled out in a bit more detail?

The methods are quite comprehensive in their scope. While quite dense, they appear to have been developed and described quite rigorously.

**Have the authors made all data and (if applicable) computational code underlying the findings in their manuscript fully available?**

Reviewer #2: **No: **I may have missed it, but it doesn't appear that they are sharing the code for their proposed tool.

PLOS authors have the option to publish the peer review history of their article (what does this mean?). If published, this will include your full peer review and any attached files.

Reviewer #1: No

Reviewer #2: No

**Have all data underlying the figures and results presented in the manuscript been provided?**

Reviewer #1: Yes

Figure Files:

Data Requirements:

Reproducibility:

References:

---

## [Editor Report · Decision Letter 1]

24 Jun 2021

Dear Mr Pospisil,

We are pleased to inform you that your manuscript 'The unbiased estimation of the fraction of variance explained by a model' has been provisionally accepted for publication in PLOS Computational Biology.

Best regards,

Frédéric E. Theunissen

Associate Editor

PLOS Computational Biology

Lyle Graham

Deputy Editor

PLOS Computational Biology

Thank you for addressing all of our concerns and sharing the code.

Best wishes,

Frederic T.

---

## [Editor Report · Acceptance letter]

23 Jul 2021

PCOMPBIOL-D-20-02116R1 

The unbiased estimation of the fraction of variance explained by a model

Dear Dr Pospisil,

I am pleased to inform you that your manuscript has been formally accepted for publication in PLOS Computational Biology. Your manuscript is now with our production department and you will be notified of the publication date in due course.

With kind regards,

Zita Barta
